# LAP: Fast LATENT DIFFUSION PLANNER WITH FINE-GRAINED FEATURE DISTILLATION FOR AUTONOMOUS DRIVING

## ABSTRACT

Diffusion models have demonstrated strong capabilities for modeling human-like driving behaviors in autonomous driving, but their iterative sampling process induces substantial latency, and operating directly on raw trajectory points forces the model to spend capacity on low-level kinematics, rather than high-level multi-modal semantics. To address these limitations, we propose **LA**tent **P**lanner (LAP), a framework that plans in a VAE-learned latent space that disentangles high-level intents from low-level kinematics, enabling our planner to capture rich, multi-modal driving strategies. We further introduce a fine-grained feature distillation mechanism to guide a better interaction and fusion between the high-level semantic planning space and the vectorized scene context. Notably, LAP can produce high-quality plans in **one single denoising step**, substantially reducing computational overhead. Through extensive evaluations on the large-scale nuPlan benchmark, LAP achieves **state-of-the-art** closed-loop performance among learning-based planning methods, while demonstrating an inference speed-up of at most $10\times$ over previous SOTA approaches. Project website: https://anonymous.4open.science/w/Latent-Planner/.

## 1 INTRODUCTION

The efficacy of modern autonomous driving systems hinges on robust motion planning, capable of navigating complex, interactive environments (Chen et al., 2024). A central challenge is handling the inherent uncertainty and behavioral multimodality of real-world traffic, where multiple distinct yet equally plausible maneuvers may be available (Yang et al., 2023; Xiao et al., 2020). While early rule-based systems offered interpretability, their hand-crafted logic is brittle and fails to scale to the long-tail of open-world scenarios (Fan et al., 2018; Chen et al., 2024). Consequently, the field has shifted towards data-driven Imitation Learning (IL), which excels at capturing nuanced, human-like behaviors from large-scale datasets (Le Mero et al., 2022; Teng et al., 2022). However, the standard IL objective is notoriously susceptible to mode-averaging, where the model collapses multiple valid expert trajectories into a single, physically infeasible path, fundamentally failing to represent the multi-modal nature of human decision-making (Strohbeck et al., 2020).

To overcome this limitation, Denoising Diffusion Probabilistic Models(DDPMs) have emerged as a powerful tool for modeling complex, multi-modal distributions (Liao et al., 2025; Ho et al., 2020). However, existing approaches models directly to raw trajectory waypoints are both computationally inefficient and conceptually flawed. This mirrors the core challenge of early image synthesis: operating in a high-dimensional pixel space expends vast model capacity on low-level details over high-level semantics (Rombach et al., 2022). A raw trajectory is analogous, as its high dimensionality is dominated by predictable kinematic redundancies (e.g., continuity, velocity limits, etc.) rather than strategic content. Consequently, training in this "waypoint space" pixel level forces the model to waste capacity on modeling basic physics, distracting from the critical task of capturing the multi-modal semantics of driving strategy.

To address this challenge, we propose **LA**tent **P**lanner (LAP), a framework that decomposes trajectory generation into two specialized stages: learning strategic semantics in a compact latent space and reconstructing high-fidelity dynamics. We first design a Variational Autoencoder (VAE) (Kingma et al., 2013) to learn a low-dimensional latent space that captures the strategic essence

of trajectories while abstracting away kinematic details. A conditional transformer-based diffusion model is then trained on these latents to focus on modeling the multi-modal distribution of high-level driving policies. Furthermore, to mitigate the modality gap between the high-level semantic planning space and the low-level vectorized input representations, we introduce a fine-grained feature distillation method to facilitate a more effective interaction and fusion of information between them.

This two-stage approach yields a dual advantage: it dramatically improves generation efficiency by confining the diffusion process to a low-dimensional latent space, while simultaneously ensuring both strategic diversity and high-fidelity, kinematically feasible outputs. Through extensive evaluations on the large-scale nuPlan benchmark (Caesar et al., 2021), we demonstrate that LAP establishes a new state-of-the-art in closed-loop performance among learning-based methods. Notably, this is accomplished with up to a $10\times$ speed-up in inference, producing high-quality plans that substantially reduce the reliance on hand-crafted post-processing.

In summary, our key contributions are:

- We propose a latent diffusion framework for autonomous planning that disentangles high-level strategic semantics from low-level kinematic execution, leading to improvements in both performance and computational efficiency.

- We design a specialized trajectory VAE that learns a compact, semantically rich latent space while ensuring high-fidelity, kinematically feasible reconstructions.

- We introduce a novel fine-grained feature distillation method to bridge the gap between the high-level semantic planning space and low-level vectorized scene perception, facilitating the information interacion between them.

- Our model, LAP, establishes a new state-of-the-art in closed-loop performance on the nuPlan benchmark for learning-based planners while demonstrating a substantial reduction in inference latency.

## 2 RELATED WORKS

Traditional motion planners are typically formulated as rule-based finite state machines with if-then-else logic (Zhou et al., 2022), a paradigm valued for its interpretability and verifiable safety guarantees (Fan et al., 2018; Urmson et al., 2008). However, the reliance on hand-crafted logic makes these systems inherently brittle and difficult to scale, as preventing conflicts between an ever-expanding set of rules becomes exponentially challenging (Grigorescu et al., 2020). This fundamental limitation makes it intractable to exhaustively cover the infinite long-tail of novel scenarios found in dense, dynamic traffic, ultimately hindering their real-world applicability (Karnchanachari et al., 2024; Lu et al., 2024).

To overcome the scalability issues of rule-based planners, Imitation Learning (IL) has become the predominant paradigm, training policies to mimic the behavioral patterns of human experts from large-scale datasets (Le Mero et al., 2022; Codevilla et al., 2018). Such data-driven approach has spurred the development of increasingly sophisticated architectures, from early RNN-based models to powerful Transformer-based frameworks that capture complex, multi-modal environmental context (Bansal et al., 2018; Chitta et al., 2022; Cheng et al., 2024). However, the standard IL objective is notoriously susceptible to mode-averaging, where multiple valid expert maneuvers are collapsed into a single and often kinematically infeasible trajectory (Strohbeck et al., 2020). This failure to capture the multi-modal nature of human decision-making motivates the shift towards generative models that can explicitly model diverse behavioral distributions.

To address the problem of mode-averaging, generative models have become an active area of research. An early, conceptually powerful approach was to perform planning in a latent space learned by a Variational Autoencoder, which could capture the semantic diversity of driving maneuvers (Zheng et al., 2024). However, these methods were often constrained by the VAE's limited generative fidelity, struggling to produce kinematically realistic trajectories.

To resolve the issue of generation quality, recent works have adopted Denoising Diffusion Models as the core of the planning policy (Chi et al., 2023). This paradigm excels at producing high-fidelity, diverse trajectories, with state-of-the-art frameworks demonstrating the ability to generate high-quality plans directly, removing the need for rule-based post-processing common in prior methods (Zheng

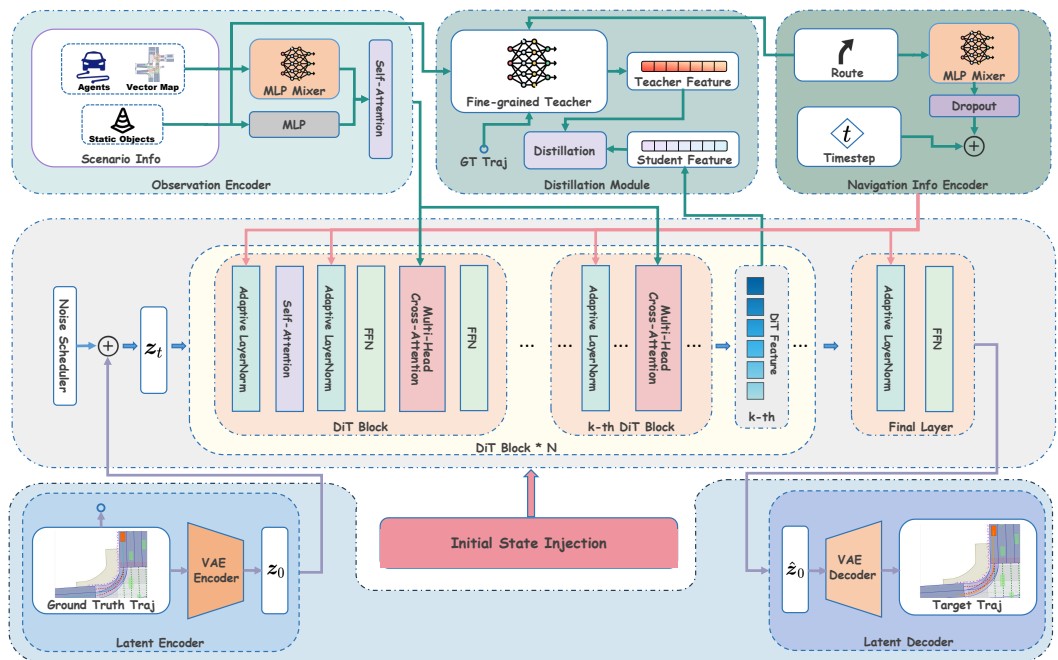

Figure 1: Overall architecture of *Latent Planner*

et al., 2025). Subsequent efforts have further improved sample diversity and efficiency (Jiang et al., 2025; Liao et al., 2025). Nevertheless, a fundamental limitation shared by all these methods is their operation directly on raw trajectory waypoints. This forces the model to expend significant capacity learning low-level kinematics, rather than focusing its power on the more critical task of high-level strategic decision-making.

## 3 PRELIMINARIES

**Diffusion Models.** Diffusion probabilistic models (Ho et al., 2020) are a class of generative models that produce samples by reversing a forward noising process. In the forward process, which can be defined over a continuous time interval $t \in [0, 1]$, noise is progressively added to a data sample $\boldsymbol{x}_0$ according to a predefined noise schedule. The distribution of the data sample at time $t$ can be expressed in a closed form as:

$$q_{t0}(\boldsymbol{x}_t|\boldsymbol{x}_0) = \mathcal{N}(\boldsymbol{x}_t; \alpha_t \boldsymbol{x}_0, \sigma_t^2 \mathbf{I}), t \in [0, 1], \tag{1}$$

where $\sigma_t^2$ is the noise variance at time $t$ and $\alpha_t := \sqrt{1 - \sigma_t^2}$. The schedule is designed such that $\alpha_t \to 0$ and $\sigma_t^2 \to 1$ as $t \to 1$, ensuring that $\boldsymbol{x}_t$ converges to a standard Gaussian distribution. During inference, the original data is recovered from noise by solving an equivalent reverse-time Diffusion ODE (Song et al., 2020) of the forward process in equation 1:

$$\frac{\mathrm{d}\boldsymbol{x}_t}{\mathrm{d}t} = f(t)\boldsymbol{x}_t - \frac{1}{2}g^2(t)\nabla_{\boldsymbol{x}_t} \log q_t(\boldsymbol{x}_t), \tag{2}$$

where $f(t) = \frac{\mathrm{d}\log \alpha_t}{\mathrm{d}t}, g^2(t) = \frac{\mathrm{d}\sigma_t^2}{\mathrm{d}t} - 2\frac{\mathrm{d}\log \alpha_t}{\mathrm{d}t}\sigma_t^2$, and $q_t$ is the marginal distribution of $\boldsymbol{x}_t$. Diffusion models utilize a neural network $\boldsymbol{s}_\theta(\boldsymbol{x}_t, t)$ to estimate the score function $\nabla_{\boldsymbol{x}_t} \log q_t(\boldsymbol{x}_t)$ (Song & Ermon, 2019), enabling the modeling of arbitrarily complex distributions.

**Classifier-free Guidance.** Classifier-free Guidance (CFG) (Ho & Salimans, 2022) is a widely used technique to enhance the alignment between generated samples and input conditions in diffusion models. The key idea is to randomly drop the conditioning information with a certain probability $p \in [0, 1)$ during training, enabling a single model to learn both conditional and unconditional generation. At the inference stage, the model predicts the score functions for both the conditional and unconditional cases. The final guidance direction is then formed by a linear combination of

these two, with the strength of the conditioning controlled by a guidance scale parameter $\omega$:

$$\tilde{s}_\theta(z_t, t, \text{cond}) = s_\theta(z_t, t, \varnothing) + \omega \left[ s_\theta(z_t, t, \text{cond}) - s_\theta(z_t, t, \varnothing) \right], \tag{3}$$

where cond is the input conditions. The parameter $\omega \geq 0$ controls the trade-off between conditional fidelity and sample diversity. Higher values of $\omega$ lead to stronger adherence to the condition, whereas lower values result in greater diversity.

## 4 METHODOLOGY

### 4.1 OVERVIEW

**Problem Formulation.** Following the problem formulation of Diffusion Planner (Zheng et al., 2025), our task is to plan the future trajectory for the ego-vehicle while jointly considering the potential future trajectories of $M$ neighbor agents, given a condition $C$. This condition includes current vehicle states, historical data, lane information, and navigation information. The target is a collection of trajectories $\mathcal{T} = \{\tau_0, \tau_1, \dots, \tau_M\}$, where $\tau_0$ is the ego-vehicle's trajectory and $\{\tau_i\}_{i=1}^M$ are the trajectories of the neighbors. Each trajectory consists of states over a future horizon of $T$ timesteps:

$$\mathcal{T} = \begin{bmatrix} \tau_0 \\ \tau_1 \\ \vdots \\ \tau_M \end{bmatrix} = \begin{bmatrix} s_0^{(0)} & s_0^{(1)} & \cdots & s_0^{(T)} \\ s_1^{(0)} & s_1^{(1)} & \cdots & s_1^{(T)} \\ \vdots & \vdots & \ddots & \vdots \\ s_M^{(0)} & s_M^{(1)} & \cdots & s_M^{(T)} \end{bmatrix} \in \mathbb{R}^{(1+M) \times (1+T) \times 4},$$

where $s_i^{(t)}$ denotes the state of agent $i$ at future timestep $t$. The state is represented by its coordinates and the sine and cosine of its heading angle.

**Overall Architecture.** Our model, illustrated in Figure 1, is a Latent Diffusion Transformer (Peebles & Xie, 2023) designed for trajectory planning. The training process consists of two stages: First, we train a Variational Autoencoder (Kingma et al., 2013) responsible for encoding trajectories into a compact latent vector $z_0$ and decoding it back to the original trajectory space. Second, we train a conditional diffusion model within the learned latent space. This model is trained to reverse a noising process, where it learns to denoise a noisy latent representation $z_t$ back to the original latent representation $z_0$, conditioned on the scene context $C$. By operating in this compressed latent space, our model can efficiently generate multi-modal trajectory plans.

### 4.2 TRAJECTORY REPRESENTATION IN LATENTS

Our proposed Trajectory Variational AutoEncoder ($\mathcal{V}$), comprising an encoder $\mathcal{E}$ and a decoder $\mathcal{D}$, is designed based on the Transformer architecture (Vaswani et al., 2017) to capture temporal dependencies within trajectory data. We augment the standard VAE objective with an auxiliary differential loss, which regularizes the model to produce smoother reconstructions. The encoder $\mathcal{E}$ compresses each trajectory $\tau_i$ into an $L$-dimensional latent space $\mathcal{Z}$, from which the decoder $\mathcal{D}$ reconstructs the trajectory.

More specifically, the encoder $\mathcal{E}$ first projects the input trajectories $\mathcal{T}$ into a sequence of high-dimensional embeddings. A set of learnable queries then interacts with these embeddings through a self-attention to aggregate the trajectories' essential features. These queries are then processed to produce the mean $\mu \in \mathbb{R}^{(1+M) \times L}$ and diagonal covariance matrix $\Sigma \in \mathbb{R}^{(1+M) \times L \times L}$ of the latent distribution. The decoder's operation begins with sampling a latent representation $z_0 \sim \mathcal{N}(\mu, \Sigma) = q(z|\mathcal{T})$ using the reparameterization trick. It then utilizes a distinct set of way-point queries, which attend to the latent representation $z_0$ via a multi-head cross attention. This process allows the way-point queries to gather the necessary information for reconstruction, after which they are projected to generate the reconstructed trajectories $\hat{\mathcal{T}}$.

To obtain a more informative latent representation, we adopt the $\beta$-VAE framework. This approach balances the compactness of the latent space with reconstruction quality by adding a weight, $\beta$, to the Kullback-Leibler (KL) divergence term of the original VAE training loss. We use the Mean Squared Error (MSE) as our reconstruction loss, which includes a trajectory reconstruction term and a differential reconstruction term. Let $\Delta$ denote the forward temporal difference operator, the final

training objective for our VAE is formulated as:

$$\mathcal{L}_{\text{vae}} = \|\mathcal{T} - \hat{\mathcal{T}}\|^2 + \lambda\|\Delta\mathcal{T} - \Delta\hat{\mathcal{T}}\|^2 + \beta D_{\text{KL}}\left[q(\boldsymbol{z}|\mathcal{T})\|p(\boldsymbol{z})\right], p(\boldsymbol{z}) = \mathcal{N}(\mathbf{0}, \mathbf{I}), \quad (4)$$

where $\lambda$ is introduced to enhance the reconstruction quality. In our experiments, we set $\lambda = 0.01$. Further details about the model, the impact of $\lambda$ on reconstruction quality, and corresponding latent space visualizations(including interpolation, clustering, and visualization via dimensionality reduction) are provided in Appendix C.

## 4.3 PLANNING ON LATENTS

After obtaining the pre-trained VAE, we train our planner, denoted as $\boldsymbol{z}_\theta(\boldsymbol{z}_t, t, \boldsymbol{C})$, in the latent space. The training objective is to recover the original latents, $\boldsymbol{z}_0$, from the noisy latents (Ramesh et al., 2022):

$$\mathcal{L}_{\text{diff}} = \mathbb{E}_\mathcal{T}\mathbb{E}_{\boldsymbol{z}_0 \sim q(\boldsymbol{z}|\mathcal{T}), t \sim \mathbb{U}(0,1), \boldsymbol{z}_t \sim q_{t0}(\boldsymbol{z}_t|\boldsymbol{z}_0)}\left[\|\boldsymbol{z}_\theta(\boldsymbol{z}_t, t, \boldsymbol{C}) - \boldsymbol{z}_0\|^2\right]. \quad (5)$$

The details of the architecture are provided below.

**Latent Diffusion Planner.** We build our Latent Planner upon Diffusion Planner (Zheng et al., 2025). The process begins by encoding the various input modalities. Specifically, we use two separate MLP-Mixers (Tolstikhin et al., 2021) to encode the historical information of neighboring vehicles $\boldsymbol{S}_{\text{neighbor}} \in \mathbb{R}^{A \times D_{\text{neighbor}}}$, and the lane segment information $\boldsymbol{S}_{\text{lane}} \in \mathbb{R}^{P \times D_{\text{lane}}}$, respectively. A standard MLP is used to encode the static obstacle information $\boldsymbol{S}_{\text{static}} \in \mathbb{R}^{D_{\text{static}}}$. Here, $A$ denotes the number of historical time steps, $P$ is the number of lane segments, and $D_{\text{neighbor}}$, $D_{\text{lane}}$, and $D_{\text{static}}$ are the respective feature dimensions. The resulting encoded features are then fed into a transformer encoder to obtain the fused feature representation $\boldsymbol{F}_{\text{scenario}}$. Finally, we fuse $\boldsymbol{F}_{\text{scenario}}$ and $\boldsymbol{z}$ using a multi-head cross attention:

$$\boldsymbol{z} = \text{MHCA}\left(Q = \boldsymbol{z}, K = V = \boldsymbol{F}_{\text{scenario}}\right). \quad (6)$$

Additionally, we incorporate navigation information denoted by $\boldsymbol{S}_{\text{route}} \in \mathbb{R}^{(K \times P) \times D_{\text{route}}}$, where $K$ is the number of route lanes and $D_{\text{route}}$ is its feature dimension. We use MLP-Mixer to encode this information, and the resulting embedding, along with the diffusion timestep $t$, is injected into the model via adaptive layer normalization.

**Initial State Injection.** We empirically observe that without additional information, the prediction model for surrounding vehicles fails to converge. This might be attributable to the inherent variability in the initial states of neighbor vehicles, which makes the prediction task more challenging compared to that of the ego-vehicle. To address this, we inject the initial state of the surrounding vehicles, $s_{\text{init}} = [s_1^{(0)}, \ldots, s_M^{(0)}]$, as a conditional prior into the denoising process(Figure 2). Specifically, we encode $s_{\text{init}}$ using an MLP and add the resulting embedding $e_{\text{init}}$ to the input of the first DiT block and the output of the final block.

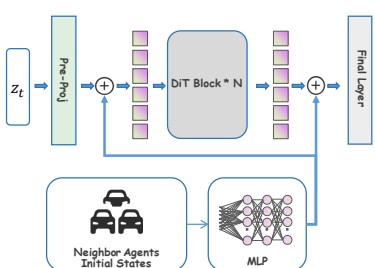

Figure 2: Initial State Injection.

This conditioning effectively anchors the prediction to a known starting point, significantly stabilizing the training process and improving convergence.

**Navigation Guidance Augmentation.** We identified a causal confusion effect in the reactive scenarios, where the ego vehicle may base its decisions entirely on the current states of surrounding vehicles. To mitigate this, one solution is to strengthen the guidance from the navigation information, effectively 'forcing' the ego-vehicle to adhere more closely to the navigation route. We employ classifier-free guidance (Ho & Salimans, 2022) to achieve this objective. Specifically, during training, we randomly drop out the navigation information with a probability $p$. This allows the model to learn both a conditional (with navigation) and an unconditional (without navigation) denoising process. At inference time, the denoising is performed using a linear interpolation of the outputs from these two modes:

$$\tilde{\boldsymbol{z}}_\theta(\boldsymbol{z}_t, t, \boldsymbol{S}_{\text{route}}) = \boldsymbol{z}_\theta(\boldsymbol{z}_t, t, \varnothing) + \omega\left[\boldsymbol{z}_\theta(\boldsymbol{z}_t, t, \boldsymbol{S}_{\text{route}}) - \boldsymbol{z}_\theta(\boldsymbol{z}_t, t, \varnothing)\right], \quad (7)$$

where $\omega$ is the guidance scale. Here, we omit the scene information $\boldsymbol{F}_{\text{scenario}}$ for simplicity. In practice, we find that $\omega = 1$ works well for most scenarios, and thus set it as the default. Then, equation 7 becomes:

$$\tilde{\boldsymbol{z}}_\theta(\boldsymbol{z}_t, t, \boldsymbol{S}_{\text{route}}) = \boldsymbol{z}_\theta(\boldsymbol{z}_t, t, \boldsymbol{S}_{\text{route}}) \quad (8)$$

This means that the forward pass of the unconditional model is no longer required, which further accelerates our model's inference speed.

### 4.4 BRIDGING THE SEMANTIC-PERCEPTION GAP

**Semantic-Perception Gap.** Planning in a compressed, low-dimensional semantic space allows for efficient learning of high-level driving policies. However, this approach creates a fundamental gap between the high-level semantic space and the low-level vectorized space of scene context. This is equivalent to converting the planning space into another modality that is completely different from the scene information, which disrupts the unified representation of agent dynamics and scene context brought by vectorized maps (Gao et al., 2020). We posit that this gap hinders effective interaction and fusion between the two spaces, leading to a performance decline compared to planners that operate directly on the pixel level.

**Fine-grained Feature Distillation.** To bridge this gap, we introduce a fine-grained feature distillation method to enhance the interaction and fusion between the planning space and the scene context. Compared to direct interaction between the high-level planning space and low-level vectorized scene context, information exchange within a unified, fine-grained vectorized space is significantly more straightforward. Therefore, we begin by encoding fine-grained trajectory-scene interaction information in the vectorized space to serve as a target feature. This feature then acts as a guiding exemplar for the intermediate layers of our DiT. Formally, let $f$ be the encoder for the trajectory $\mathcal{T}$ and the condition $\boldsymbol{C}$. Let $\boldsymbol{y}^* = f(\mathcal{T}, \boldsymbol{C}) \in \mathbb{R}^{(1+M) \times D}$ be the target feature and $\boldsymbol{h}_k \in \mathbb{R}^{(1+M) \times D}$ be the intermediate feature from the $k$-th layer of DiT, where $D$ is the embedding dimension. We augment the training objective with the following additional regularization term:

$$\mathcal{L}_{\text{dist}} = \mathbb{E}_{\mathcal{T}} \mathbb{E}_{\boldsymbol{z}_0 \sim q(\boldsymbol{z}|\mathcal{T}), t \sim \mathbb{U}(0,1), \boldsymbol{z}_t \sim q_{t0}(\boldsymbol{z}_t|\boldsymbol{z}_0)} \left[ \|h_\phi(\boldsymbol{h}_k) - \boldsymbol{y}^*\|^2 \right], \tag{9}$$

where $h_\phi$ is a learnable MLP projection head. The training objective becomes:

$$\mathcal{L} = \mathcal{L}_{\text{diff}} + \alpha \mathcal{L}_{\text{dist}}, \tag{10}$$

where $\alpha > 0$ is a hyperparameter that controls the tradeoff between denoising and distillation. The core idea is to provide the intermediate features of our DiT with a pre-encoded exemplar to imitate. This exemplar captures the fine-grained, pixel-level interaction between the vectorized planning behavior and the scene. By doing so, we offer direct guidance for fusing the high-level semantic planning space with the vectorized scene context, thereby significantly simplifying their interaction process. This distilled, fine-grained feature acts as a conceptual bridge, effectively closing the gap between the high-level planning space and the low-level vectorized scene representation.

**Implementation.** Building on the finding from Agarwal et al. (2025) that pixel-level diffusion models are effective encoders for fine-grained features and condition-image alignment, we leverage the pixel-level predictor, Diffusion Planner (Zheng et al., 2025), as the encoder $f$. The target feature $\boldsymbol{y}^*$ is then defined as the activations from its final DiT block, which takes the clean trajectory $\mathcal{T}$ and the condition $\boldsymbol{C}$ as input. This strategy effectively distills the geometric understanding of an expert pixel-level planner into our latent planning framework, ensuring the learned latents are deeply grounded in the vectorized scene context. Please refer to Appendix H for more details of the feature distillation module.

## 5 EXPERIMENTS

**Implementation Details.** We employ the same data augmentation techniques as Diffusion Planner (Zheng et al., 2025) to enhance the model's generalization to out-of-distribution (OOD) scenarios in closed-loop evaluations. Following previous work (Huang et al., 2023) , we apply z-normalization to the original trajectories. Additionally, we scale the target latents $\boldsymbol{z}_0$ by the inverse of their global standard deviation to stabilize the training process (Rombach et al., 2022). At inference time, we employ the DPM-Solver (Lu et al., 2022) for accelerated sampling. Benefiting from the compactness and smoothness of the latent space, our model can perform rapid inference within two steps. We provide a quantitative analysis of the few-step sampling in Appendix G. For further details on training and inference, please refer to Appendix D.

**Evaluation Setup**. We train and evaluate our model, LAP, on the nuPlan benchmark (Caesar et al., 2021). nuPlan is a large-scale closed-loop autonomous driving simulation framework built upon 1,300 hours of real-world driving logs, encompassing 75 distinct, pre-classified scenarios. Its simulation models traffic agents as either Non-Reactive (replaying logs) or Reactive (governed by an

Table 1: Closed-loop planning results on nuPlan dataset. For each group, the best and second-best results are highlighted in **bold** and underline, respectively. *: Using pre-searched reference lines as model input provides prior knowledge, reducing the difficulty of planning compared to standard learning-based methods.

| Type | Planner | Non-Reactive | | | Reactive | | |
|------|---------|------|------|------|------|------|------|
| | | **Val14** | **Test14-hard** | **Test14** | **Val14** | **Test14-hard** | **Test14** |
| Expert | Log-Replay | 93.53 | 85.96 | 94.03 | 80.32 | 68.80 | 75.86 |
| Rule-based & Hybrid | IDM | 75.60 | 56.15 | 70.39 | 77.33 | 62.26 | 74.42 |
| | PDM-Closed | 92.84 | 65.08 | 90.05 | 92.12 | 75.19 | 91.63 |
| | PDM-Hybrid | 92.77 | 65.99 | 90.10 | 92.11 | 76.07 | 91.28 |
| | PLUTO | 93.17 | **80.08** | 92.60 | 90.78 | 76.88 | 91.65 |
| | Diffusion Planner w/ refine. | 94.24 | 78.91 | 94.13 | **92.76** | **81.66** | 91.85 |
| | LAP w/ refine. ($o1s1$,Ours) | 94.7 | 78.26 | 93.9 | 91.85 | 78.97 | **92.35** |
| | LAP w/ refine. ($o1s2$,Ours) | **95.03** | 78.06 | **94.4** | 91.95 | 79.89 | 92.2 |
| Learning-based | PDM-Open* | 53.53 | 33.51 | 52.81 | 54.24 | 35.83 | 57.23 |
| | UrbanDriver | 53.05 | 43.89 | 51.83 | 50.42 | 42.26 | 52.34 |
| | GC-PGP | 59.51 | 46.13 | 61.16 | 55.30 | 42.74 | 54.86 |
| | PLUTO w/o refine.* | 88.89 | 74.39 | 90.11 | 78.92 | 59.30 | 80.37 |
| | Diffusion Planner | **89.64** | 75.44 | 88.84 | **82.80** | 68.95 | 82.61 |
| | LAP ($o1s1$,Ours) | 89.15 | 78.11 | 89.85 | 81.82 | 69.05 | 84.24 |
| | LAP ($o1s2$,Ours) | 89.37 | **78.52** | **90.42** | 82.23 | **70.49** | **85.12** |

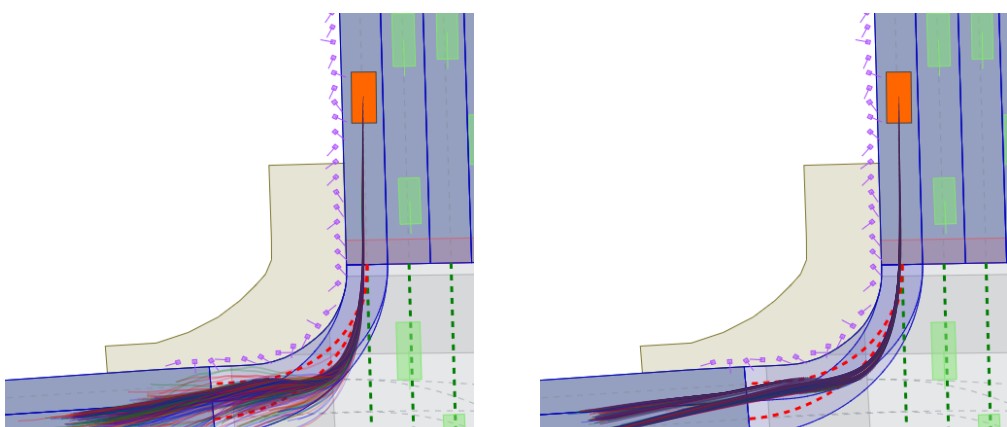

Figure 3: **Comparison of trajectory proposals for a right turn scenario.** This figure illustrates the behavior of LAP (left) and Diffusion Planner (right) , which samples $K_{\text{mode}} = 3$ candidate trajectories (thin colored lines) at each planning cycle. Notably, the proposals from our LAP planner exhibit significant multi-modality, covering a diverse range of turning radii and speeds while proceeding along the navigation route.

IDM policy (Treiber et al., 2000)). For the SOTA comparison, we use a larger model with 192 hidden dimensions and 6 attention heads. In the ablation studies, excluding the one on sampling methods, we validate our approach with a smaller model with 128 hidden dimensions and 4 attention heads. Throughout the experiments, $o$ signifies the solver order and $s$ indicates the denoising steps.

**Baselines**. The baseline methods are categorized into three groups (Dauner et al., 2023): Rule-based, Learning-based, and Hybrid. Hybrid approaches typically augment learning-based outputs with a refinement stage. To ensure a fair comparison, we apply a common post-processing refinement module (Sun et al., 2025) to our model's output without parameter tuning.

- **IDM** (Treiber et al., 2000): A classic rule-based car-following planner implemented by nuPlan.

- **PDM** (Dauner et al., 2023): The nuPlan challenge winner, with rule-based (**PDM-Closed**), learning-based (**PDM-Open**), and hybrid (**PDM-Hybrid**) versions.

- **GC-PGP** (Hallgarten et al., 2023): A learning-based predictive model designed for goal-oriented navigation on lane graphs.

- **UrbanDriver** (Scheel et al., 2021): A learning-based planner using policy gradient optimization.
- **PLUTO** (Cheng et al., 2024): An enhanced version of **PDM-Open** that uses contrastive learning for improved scene understanding.
- **Diffusion Planner** (Zheng et al., 2025): A specifically designed transformer-based diffusion model for high performance motion planning.

**Evaluation Metrics**. The nuPlan benchmark is derived from the 14 scenario categories, with each category containing up to 100 distinct scenes.The nuPlan framework provides three primary evaluation scores: an Open-Loop Score (OLS), a Non-Reactive Closed-Loop Score (NR-score), and a Reactive Closed-Loop Score (R-score). Consistent with prior research indicating a weak correlation between open-loop prediction and closed-loop planning effectiveness(Chen et al., 2024), we focus exclusively on the closed-loop scores. The closed-loop score comprises a weighted average of several key metrics, as discussed further in the Appendix F.

## 5.1 Main Results

The comparison results with state-of-the-art methods on the nuPlan dataset are presented in Table 1. Compared to all learning-based baselines, LAP achieves SOTA performance on most benchmarks within just two denoising steps. Notably, on the challenging Test14 hard dataset, our method shows a significant performance advantage over other learning-based

Table 2: Inference latency test

| Planner | Inference time (ms) ↓ | score ↑ | GFLOPS | model size(M) |
|---|---|---|---|---|
| UrbanDriver | 67.47 | 43.89 | - | - |
| GC-PGP | 39.24 | 46.13 | - | - |
| Pluto w/o refine | 745.87 | 73.61 | **0.994** | **4.24** |
| DiffusionPlanner | 202.60 | 75.44 | 1.38 | 6.04 |
| Latent Planner($o1s1$) | **18.81** | 78.11 | 1.30 | 7.03 |
| Latent Planner($o1s2$) | 21.69 | **78.52** | 1.33 | 7.03 |

methods, indicating its capability to make better decisions in complex environments. With the addition of PDM-based post-processing module, LAP's performance improves on most benchmarks, becoming comparable to SOTA rule-based and hybrid methods, and even surpassing human performance. Interestingly, on the Test14-hard NR benchmark, the performance of LAP shows a slight degradation after post-refinement. This suggests that in complex environments, LAP's decisions may already be superior to the PDM scoring module.

We also evaluated the inference latency of the SOTA learning-based methods, and the results are shown in Table 2. Although LAP and Diffusion Planner (Zheng et al., 2025) have comparable GFLOPs(most GFLOPs are spent in the scene encoder), LAP benefits from few-step sampling, so most of its computation is executed in parallel on the GPU with greatly reduced serial steps, resulting in up to a 10× improvement in inference speed. Detailed information regarding the inference speed test can be found in Appendix D.

Another key advantage of planning in the latent semantic space is the ability to capture diverse, high-level driving strategies. This is demonstrated in Figure 3, which visualizes the multi-modal trajectory proposals generated by LAP and Diffusion Planner (Zheng et al., 2025) for a right-turn scenario.

## 5.2 Latent Planning vs. Pixel-level Planning

To make the advantages of planning in the latent space more intuitive, we summarize in Table 3 a comparison of its performance on the challenging Test14-hard NR benchmark, its capacity to model multi-modal trajectories, and its inference latency. Benefiting from the VAE, which compresses raw trajectories into high-level semantic representations, planning in the latent space can handle challenging scenarios more effectively while requiring less inference time. At the same time, planning in the semantic space also enables the planner to better model multi-modal driving strategies. Further details about multi-modality behavior test is presented in Appendix E.

## 5.3 Ablation Studies

**Effect of Designed Modules.** Table 4 shows the effectiveness of our proposed modules, where ISI, Dist, and CFG denote *Initial State Injection*, *Fine-grained Feature Distillation*, and *Navigation Guidance Augmentation*, respectively. By injecting the initial state of surrounding vehicles, the model's performance is enhanced in the Non-Reactive environment but shows slight degradation in

Table 3: **Latent Planning vs. Pixel-level Planning.** APD: Average Pairwise Distance, FPD: Final Pairwise Distance. *: For a fair comparison of multi-modality, we set the decoding temperature and the number of decoding steps of both planners to 1.0 and 10, respectively.

| Planner | Test14-hard(NR) | Inference time (ms) | APD*(m) | FPD*(m) |
|---|---|---|---|---|
| Diffusion Planner | 75.44 | 202.60 | 0.88 | 1.98 |
| Latent Planner | 78.52 | 21.69 | 2.03 | 4.55 |

Table 4: Impact of designed modules.

| Modules | | | Test14-hard | |
|---|---|---|---|---|
| ISI | Dist | CFG | NR | R |
| ✗ | ✗ | ✗ | 72.03 | 66.89 |
| ✓ | ✗ | ✗ | 74.13 | 66.31 |
| ✓ | ✓ | ✗ | **76.49** | 68.89 |
| ✓ | ✓ | ✓ | 75.91 | **70.36** |

Table 5: Impact of different sampling methods.

| Sampling Method | NFE | Test14-hard | |
|---|---|---|---|
| | | NR | R |
| $o1s1$ | 1 | 78.11 | 69.05 |
| $o1s2$ | 2 | 78.52 | 70.49 |
| $o1s3$ | 3 | 78.48 | 69.91 |
| $o1s10$ | 10 | 78.33 | 69.21 |
| $o2s1$ | 2 | 78.17 | 69.93 |
| $o2s2$ | 4 | 78.23 | 70.45 |
| $o2s3$ | 6 | 77.46 | 69.53 |
| $o2s10$ | 20 | 77.13 | 69.19 |

Table 6: **Analysis of distillation strategies** on Test14-hard dataset. Navigation guidance augmentation is not applied. All metrics are measured with the DPM-Solver-1 (Lu et al., 2025) sampler with 2 denoising steps.

| Dist. Weight | Objective | Stu. Index | Test14-hard | |
|---|---|---|---|---|
| | | | NR | R |
| - | - | - | 74.13 | 66.31 |
| 0.5 | $L_2$ | 3 | 76.38 | 68.28 |
| 0.75 | $L_2$ | 3 | 76.49 | 68.89 |
| 1 | $L_2$ | 3 | 76.65 | 68.91 |
| 0.5 | $L_2$ | 2 | 75.23 | 67.48 |
| 0.75 | $L_2$ | 2 | 75.32 | 70.21 |
| 1 | $L_2$ | 2 | 75.06 | 66.72 |
| 0.5 | $L_2$ | 3 | 76.38 | 68.28 |
| 0.5 | cos.sim. | 3 | 75.98 | 65.08 |
| 0.5 | $L_2$ | 3 | 76.38 | 68.28 |
| 0.5 | $L_2$ | 2 | 75.32 | 70.21 |
| 0.5 | $L_2$ | 1 | 76.64 | 68.17 |

the Reactive environment due to causal confusion. With the further addition of the distillation term, performance in both environments improves significantly. Finally, the integration of CFG greatly alleviates the causal confusion problem and boosts performance in the reactive environment, at the cost of a marginal performance drop in the Non-Reactive environment.

**Different Sampling Methods.** Table 5 shows the impact of different sampling methods, where NFE denotes *Number of Function Evaluations*. Thanks to the latent space-based design, our model achieves excellent performance with just one denoising step, and two steps bring further improvement. However, performance degrades after three denoising steps. This is possibly due to excessively high decoding precision reducing the "flexibility" of the trajectories, which leads to a performance drop. This indicates that the latent space structure and the sampling method affect the model's performance jointly.

**Distillation Strategies.** Table 6 shows the impact of different distillation strategies, including varying the weights $\alpha$, objective functions, and the DiT layer index $k$ from which the student features are obtained. As can be seen, taking features from the final layer of the DiT as students provides a more significant and stable performance improvement compared to intermediate layers. Furthermore, weights $\alpha$ within the $[0.5, 1]$ range are beneficial for model performance. Compared to the standard $L2$ loss, relying solely on a cosine similarity constraint is insufficient and, in fact, leads to performance degradation in the Reactive environment.

**Impact of Different Teachers.** Table 7 shows the impact on the model of using different distillation targets. We compare target features obtained from various intermediate layers of the Teacher DiT and those from different magnitudes of noise adding to the ground truth trajectory. Features from the Teacher DiT's first and last layers provide a more significant improvement than those from intermediate layers. Meanwhile, features obtained by encoding original trajectories with varying degrees of added noise consistently enhance model performance.

**Impact of Different VAEs.** Table 8 shows the model performance with different VAEs, where $L$ denotes the latent dimension. It can be observed that the structure of the latent space is critically important for model performance. Although training the VAE for 200 epochs reduces its reconstruction error, it degrades the final performance of the model. This indicates that when planning in the latent space, a trade-off between its compactness and reconstruction fidelity is essential.

| Teacher Feat. | Test14-hard NR | Test14-hard R |
|---|---|---|
| Layer3 | 76.38 | 68.28 |
| Layer2 | 74.05 | 65.92 |
| Layer1 | 76.63 | 67.67 |
| Half Noise | 76.61 | 66.8 |
| Full Noise | 75.9 | 68.57 |

Table 7: **Impact of different teachers.**

| VAE | Test14-hard NR | Test14-hard R |
|---|---|---|
| L10Epo120 | 76.38 | 68.28 |
| L10Epo200 | 74.72 | 66.69 |
| L5Epo120 | 74.41 | 63.4 |
| L5Epo200 | 72.76 | 63 |

Table 8: **Impact of different VAEs.**

## 5.4 DISCUSSION ON FEATURE DISTILLATION MODULE

The above results provide a clearer picture of the role of feature distillation module in LAP. First, all planning decisions are made exclusively in the latent semantic space learned by the trajectory VAE; the teacher branch and the associated distillation loss are only used during training as an auxiliary regularizer and are removed at inference time. Second, the ablations on different distillation strategies and target features (Tables 6 and 7) suggest that the distillation loss mainly shapes better intermediate representations and strengthens the interaction between the high-level planning space and the low-level scene, in line with recent findings in the image generation literature (Yu et al., 2024). Moreover, we find that simply reintroducing pixel-level trajectory supervision is not sufficient: replacing the distillation loss with a pixel-level trajectory prediction loss degrades the model's final performance (see Appendix H for details). This suggests that the distilled teacher features must encode meaningful trajectory–scene interactions in order to effectively bridge the high-level planning space and the vectorized scene representation, rather than merely injecting low-level kinematic signals.

## 6 CONCLUSION

We introduce LAP, a latent diffusion framework that improves planning performance and efficiency by operating in a disentangled semantic space learned by a Trajectory VAE. By incorporating a novel fine-grained feature distillation method, LAP establishes a new state-of-the-art on the nuPlan benchmark while accelerating inference by up to $10\times$. For limitations and future work, see Appendix J.

## 7 REPRODUCIBILITY STATEMENT

Our work is reproducible. We provide detailed information for reproduction in Appendix D.

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

## A    THE USE OF LARGE LANGUAGE MODELS

In adherence to the conference guidelines, we use Google's Gemini 2.5 Pro, a Large Language Model (LLM), as a writing assistance tool during the preparation of this manuscript. The model was employed exclusively as an assistive tool for language and style enhancement. Its contributions were limited to tasks such as correcting grammar and syntax, improving sentence clarity and suggesting more precise academic vocabulary. The LLM also assisted in proofreading for typographical errors. The core research ideas, experimental results, and scientific conclusions presented herein are entirely the work of the human authors. We have thoroughly reviewed and edited all text and assume full responsibility for the entire content and integrity of this manuscript.

## B    VISUALIZATION OF CLOSED-LOOP PLANNING RESULTS

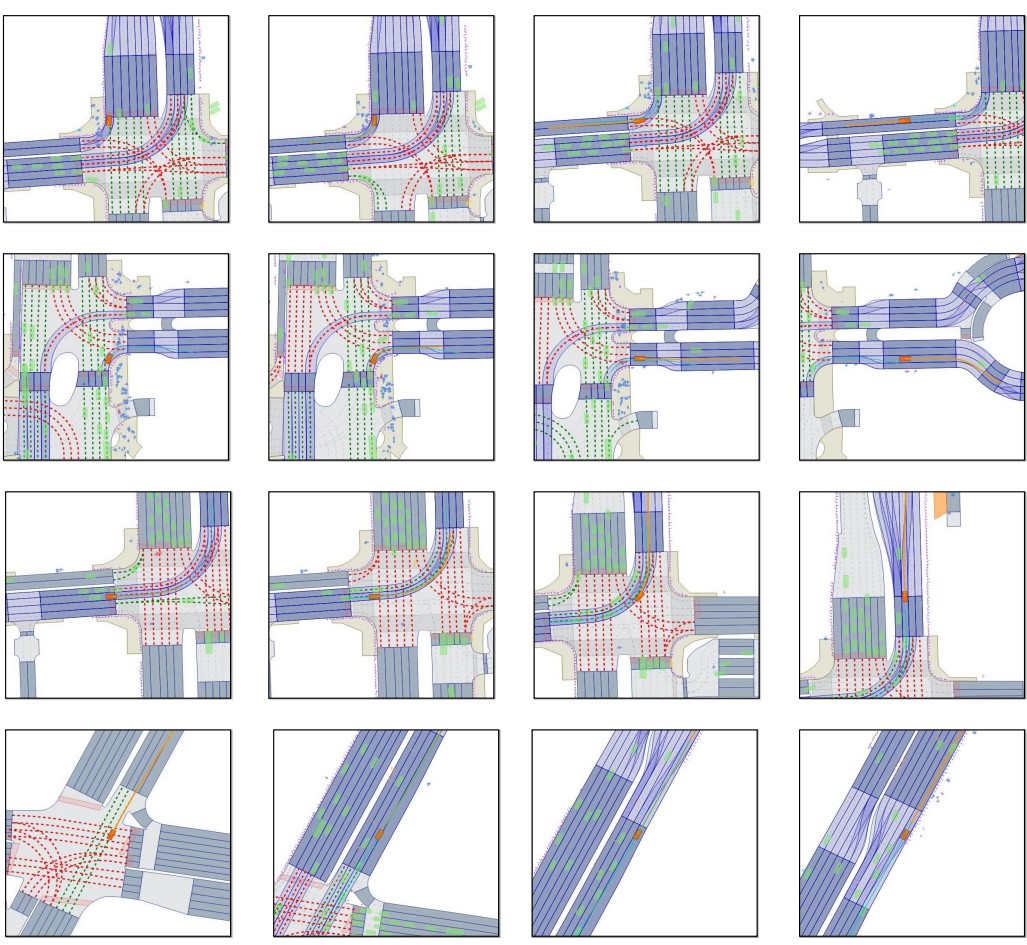

Figure 4: **Closed-loop planning results**: To showcase our planner, we have chosen 4 scenarios that involve turning, lane changing, and interactions with Vulnerable Road Users (VRUs). Each row represents a scenario at 0, 5, 10, and 15 seconds intervals. Each frame includes the future planning of the ego vehicle and the ground truth ego trajectory.

## C    IMPLEMENTATION DETAILS OF TRAJECTORY VAE

In this section, we provide the detailed architecture of our Trajectory VAE, analyze the impact of the differential loss term on reconstruction quality, and visualize the smoothness of the latent space. The details are provided below.

## C.1 DETAILED ARCHITECTURE OF TRAJECTORY VAE

**Trajectory Encoder.** Inspired by (Devlin et al., 2019; Dosovitskiy et al., 2020), we utilize learnable queries to aggregate structural information from trajectories. In the first step, we project the input trajectories into a high-dimensional space to better model their non-linear structures:

$$\mathcal{T}^{\text{proj}} = \text{MLP}(\mathcal{T}) \in \mathbb{R}^{(1+M)\times(1+T)\times H}, \tag{11}$$

where $H$ is the hidden size of our model. We then prepend a set of encoding queries $\boldsymbol{Q}^{\text{enc}} \in \mathbb{R}^{(1+M)\times N \times H}$, to the prefix of each trajectory representation and feed them into a multi-layer self-attention. The resulting query features are flattened and fed into an MLP, which project the aggregated features into a $L$-dimensional($L \ll (1+T)\times 4$) latent space for each trajectory. To enhance modeling flexibility, we employ distinct sets of queries for the ego and neighbor trajectories:

$$\boldsymbol{Q}^{\text{enc}} = \text{concat}\left(\boldsymbol{Q}^{\text{enc}}_{\text{ego}}, [\boldsymbol{Q}^{\text{enc}}_{\text{neighbor}}] \times M\right), \tag{12}$$

$$\boldsymbol{Q}^{\text{enc}}, \mathcal{T}^{\text{proj}} = \text{MHSA}\left(\text{concat}(\boldsymbol{Q}^{\text{enc}}, \mathcal{T}^{\text{proj}})\right), \tag{13}$$

$$\boldsymbol{\mu}, \boldsymbol{\sigma} = \text{MLP}(\boldsymbol{Q}^{\text{enc}}), \tag{14}$$

where $\boldsymbol{\mu}, \boldsymbol{\sigma} \in \mathbb{R}^{(1+M)\times L}$ represent the mean and standard deviation, respectively, of the latent Gaussian distribution for each trajectory. The covariance matrix $\boldsymbol{\Sigma}$ is given by:

$$\boldsymbol{\Sigma} = \text{diag}\left(\boldsymbol{\sigma}^2\right). \tag{15}$$

The high-level semantic representation is sampled using the reparameterization trick $\boldsymbol{z}_0 = \boldsymbol{\mu} + \boldsymbol{\sigma} \odot \boldsymbol{\epsilon}$.

**Trajectory Decoder.** In the decoder, a latent trajectory code $\boldsymbol{z}_0$ is first transformed by an MLP into $C$ conditional reconstruction features $\boldsymbol{C}^{\text{recon}} \in \mathbb{R}^{(1+M)\times C \times H}$. We then use a set of way-point queries (Li & Cui, 2024)$\boldsymbol{Q}^{\text{dec}} \in \mathbb{R}^{(1+M)\times(1+T)\times H}$ to extract reconstruction features, which are subsequently processed by a final MLP to yield the reconstructed trajectories $\hat{\mathcal{T}}$. Symmetric to the encoder, we employ distinct queries for the ego and neighbor agents to maintain architectural symmetry:

$$\boldsymbol{C}^{\text{recon}} = \text{MLP}(\boldsymbol{z}_0), \tag{16}$$

$$\boldsymbol{Q}^{\text{dec}} = \text{concat}\left(\boldsymbol{Q}^{\text{dec}}_{\text{ego}}, [\boldsymbol{Q}^{\text{dec}}_{\text{neighbor}}] \times M\right), \tag{17}$$

$$\hat{\mathcal{T}} = \text{MLP}\left(\text{MHCA}\left(Q = \boldsymbol{Q}^{\text{dec}}, K = V = \boldsymbol{C}^{\text{recon}}\right)\right). \tag{18}$$

## C.2 TRAINING DETAILS

**Datasets.** We randomly sample 800,000 trajectories from the nuplan dataset as our training set, 50,000 as the validation set, and 1,000 as the test set. Each data sample consists of the 8-second future trajectories for the ego vehicle and up to 32 surrounding agents. Invalid waypoints are padded with zeros.

**Training.** For the trajectory data, we employ the same data augmentation and z-score normalization techniques as used in Diffusion Planner (Zheng et al., 2025). The regularization weight $\beta$ for the KL divergence term in equation 4 is set to $1e^{-6}$. We trained the model on two NVIDIA RTX 5880 GPUs with a batch size of 256, with 2-step gradient accumulation. The model is optimized using the Adam optimizer with a learning rate of $1e^{-6}$. More hyperparameters of our VAE are provided in Table 10.

## C.3 EXPERIMENTAL RESULTS

The reconstruction quality of the VAE sets the performance upper bound for the Latent Diffusion Model (LDM). To improve reconstruction fidelity, we augment our VAE reconstruction objective with a differential loss term. To validate the effectiveness of this approach, we conducted experiments under various parameter settings. Specifically, we investigated the effects of varying the differential weight $\lambda$, the number of training epochs, and the latent dimension $L$. The results are summarized in Table 9, where we use the Average Displacement Error for both the ego agent (Ego-ADE) and its neighbors (Neighbor-ADE) as the reconstruction evaluation metrics:

$$\text{ADE}(\hat{\tau}_i, \tau_i) = \frac{1}{1+T}\sum_{k=0}^{T}\|\hat{p}_i^{(k)} - p_i^{(k)}\|_2, i = 0, 1, \cdots, M, \tag{19}$$

where $\hat{p}_i^{(k)}, p_i^{(k)}$ represent the reconstructed and original trajectory points, respectively, for the $i$-th vehicle at time step $k$. The angular reconstruction errors for all experiments are negligible (on the order of $1e^{-6}$ to $1e^{-3}$) and are therefore not reported.

We can observe that in most experimental settings, a non-zero $\lambda$ significantly improves reconstruction performance compared to the baseline where $\lambda = 0$, which demonstrates the effectiveness of the differential loss term in equation 4.

Table 9: Analysis of the differential weight $\lambda$ on reconstruction performance across different model configurations. For each hyperparameter group, the best and second-best results are highlighted in **bold** and underline, respectively.

| Latent dimension | Training Epoch | $\lambda$ | Ego-ADE(m) | Neighbor-ADE(m) |
|---|---|---|---|---|
| $L = 5$ | 120 | 0 | 0.476 | 0.868 |
| | | 0.01 | 0.320 | 0.819 |
| | | 0.1 | **0.307** | **0.777** |
| | | 1 | 0.317 | 0.786 |
| | 200 | 0 | 0.367 | 0.755 |
| | | 0.01 | **0.259** | 0.748 |
| | | 0.1 | 0.286 | **0.710** |
| | | 1 | 0.292 | 0.724 |
| $L = 10$ | 120 | 0 | 0.224 | 0.650 |
| | | 0.01 | 0.219 | 0.643 |
| | | 0.1 | 0.242 | 0.595 |
| | | 1 | **0.206** | **0.572** |
| | 200 | 0 | 0.216 | 0.560 |
| | | 0.01 | 0.170 | **0.499** |
| | | 0.1 | 0.200 | 0.590 |
| | | 1 | **0.167** | 0.506 |

## C.4 VISUALIZATION OF THE LEARNED LATENT SPACE

A compact and meaningful latent space is crucial for the performance of a LDM. In this section, we provide a visual analysis of the smoothness and semantic disentanglement of the VAE latent space, with hyperparameters set to $\lambda = 0.01$ and 120 training epochs.

### C.4.1 LATENT SPACE INTERPOLATION

To visualize the smoothness of the latent space, we randomly sample two trajectory instances from the test set and encode their ego-vehicle trajectories to obtain two latent representations, $z_1^*$ and $z_2^*$. Subsequently, we perform linear interpolation between $z_1^*$ and $z_2^*$ to generate a sequence of latent variables $[z_1, ..., z_B]$, where $z_1 = z_1^*$ and $z_B = z_2^*$. Finally, this sequence is decoded back into the trajectory space, yielding a series of interpolated trajectories. The results are presented in Figure 5, where the interpolation coefficient increases from left to right. The results demonstrate a smooth and plausible evolution from the starting trajectory to the ending one, which validates that our VAE has learned a compact and informative latent representation.

### C.4.2 LATENT SPACE CLUSTERING

To verify whether the latent space exhibits a meaningful semantic structure, we randomly sample 100k ego-vehicle trajectories and encode them with the VAE, using the corresponding mean vector as the latent representation of each trajectory. We then perform $k$-means($k = 10$) clustering on these 100k latent vectors and decode the resulting $k$ cluster centers back into the trajectory space, obtaining the prototypes shown in Fig. 6. As can be seen, the decoded prototype trajectories correspond to clearly distinct high-level maneuvers (e.g., going straight with different velocities, strong braking,

and left/right turning behaviors). This indicates that trajectories are organized in the latent space according to strategic driving intent rather than low-level waypoint coordinates.

### C.4.3 Latent Space Semantic Category Visualization

To visually assess whether the latent space captures distinct intent categories of trajectories, we perform dimensionality reduction and color the latent codes according to their intent category. Since the trajectories do not come with explicit intent labels, we first sample 100k trajectories from the original dataset and run $k$-means($k = 10$) clustering on them (Fig. 7), using the resulting cluster assignments as pseudo intent labels. We then randomly sample another 100k trajectories, project their latent codes into 2D using UMAP (McInnes et al., 2018), and color each point by its corresponding intent category, obtaining the visualization shown in Fig. 8. We observe that trajectories sharing the same intent form compact and mostly non-overlapping regions in the latent space, whereas different intents occupy well-separated areas. This demonstrates that the learned latent codes are well aligned with high-level semantic modes of driving.

However, a subset of outlier points (the purple region) is particularly noticeable, being scattered over roughly three separate areas, which we hypothesize may correspond to a special intent. To further analyze the intent represented by these outliers, we perform $k$-means clustering ($k = 3$) on this subset alone and decode the resulting cluster centers back into the original trajectory space, obtaining the trajectories shown in Fig. 9. As can be seen, all three decoded trajectories remain near the origin and almost stationary, indicating that this subset of outliers corresponds to the special intent of staying still.

## D Experimental Details

In this section, we provide the necessary details to reproduce our work.

### D.1 Training Details

We use the same training data as Diffusion Planner (Zheng et al., 2025), which consists of 1 million scenarios randomly sampled from the nuPlan training data. The model was trained for 200 epochs on 4 NVIDIA RTX 5880 (48GB) GPUs with a batch size of 1024. We employed the AdamW optimizer (Loshchilov & Hutter, 2017) with a learning rate of $5e^{-4}$, preceded by a 5-epoch linear warmup phase. A detailed summary of the setup is reported in Table 10.

### D.2 Inference Details

We utilize DPM-Solver++ as the solver for the reverse diffusion process and adopt a variance-preserving (VP) noise schedule:

$$\sigma_t = (1 - t)\beta_{\min} + t\beta_{\max}. \tag{20}$$

We set the initial sampling temperature to 1.0 and used a default guidance scale of 1.0 throughout all experiments, with further hyperparameter details also provided in Table 10.

### D.3 Inference Time Test Details.

We tested the inference latency of our proposed Latent Planner and all baseline methods under an identical experimental setup. The evaluation was conducted by running 10,000 scenarios from the challenging Test14-Hard Non-reactive dataset on a single NVIDIA RTX 5880 GPU (48GB) using the official nuPlan simulation framework. All baseline methods were tested using their official codebase to ensure a fair comparison.

### D.4 Baseline Setup

**nuPlan Datasets Evaluation**. For the *IDM*, *UrbanDriver*, and *GC-PGP* baselines, we utilize the official nuPlan codebase[1]. The checkpoints for both UrbanDriver and GC-PGP are sourced from

---

[1] https://github.com/motional/nuplan-devkit

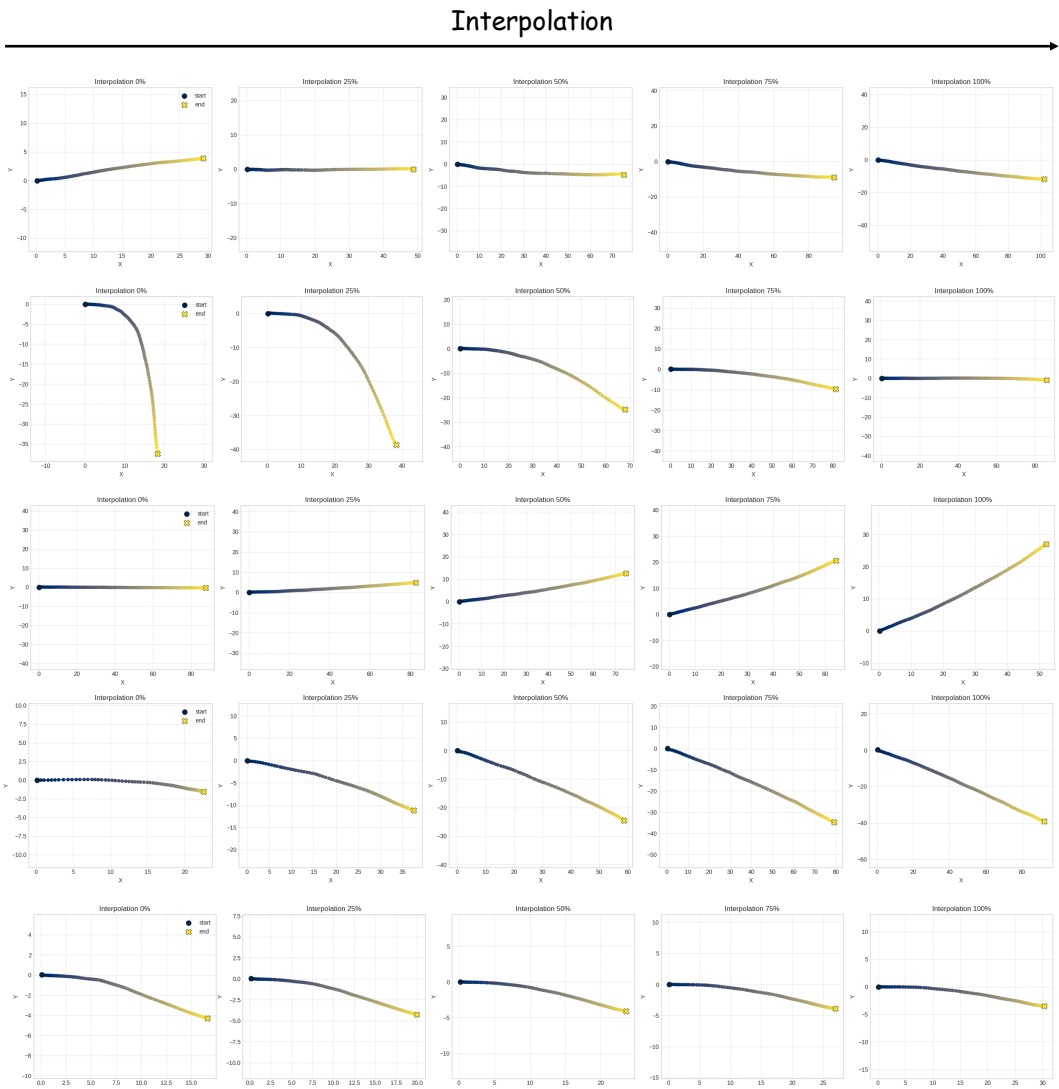

Figure 5: **Smooth Trajectory Transitions via Latent Space Interpolation.** The figure displays five examples of decoded trajectories generated by linearly interpolating between the latent codes of two real trajectories. The smooth and plausible nature of the intermediate trajectories (with $B = 5$ steps) demonstrates that our model has learned a continuous and meaningful latent representation.

the PDM codebase[2], which also provides the checkpoints for *PDM-Hybrid* and *PDM-Open*. For *PLUTO* and *Diffusion Planner*, we directly employ the checkpoints from their respective official codebases[3][4]. For *PLUTO w/o refine*, we select the most probable trajectory from the model output, skip post-processing, and rerun the simulation without retraining. For *Diffusion Planner w/ refine*, we incorporate the post-processing module from the STR codebase[5], as suggested from *Diffusion Planner*, which is consistent with the post-processing approach used in *Latent Planner w/ refine*.

---

[2]https://github.com/autonomousvision/tuplan_garage

[3]https://github.com/jchengai/pluto

[4]https://github.com/ZhengYinan-AIR/Diffusion-Planner

[5]https://github.com/Tsinghua-MARS-Lab/StateTransformer

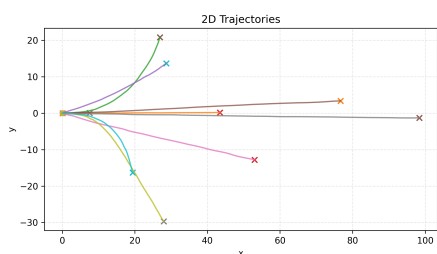

Figure 6: Latent space clustering.

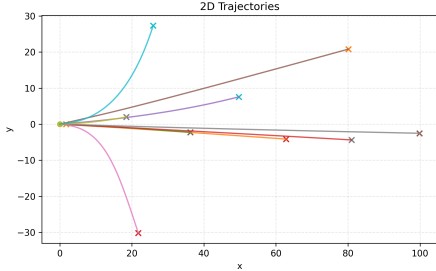

Figure 7: Trajectory space clustering.

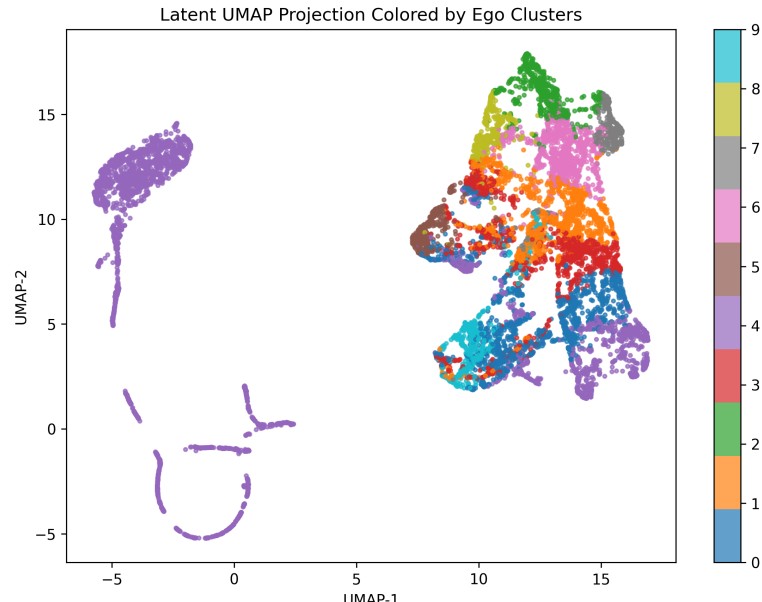

Figure 8: UMAP visualization of the latent space.

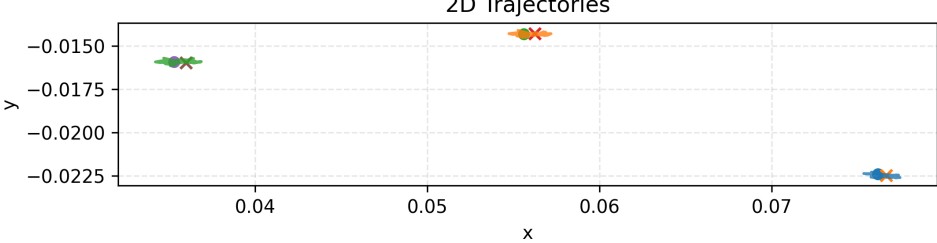

Figure 9: Outlier analysis.

# E    MULTI-MODAL PLANNING BEHAVIOR

To demonstrate the advantage of our method, LAP, in generating diverse trajectories, we conduct a comparative evaluation against Diffusion Planner (Zheng et al., 2025) following the evaluation method in DLow (Yuan & Kitani, 2020). For this, we sample $K_{\text{mode}}$ trajectories and compute the **Average Pairwise Distance** and **Final Pairwise Distance** over them. These two metrics measure the average spatial separation between all possible pairs of trajectories, where

Table 10: Hyperparameters of *Latent Planner*

| Type | Parameter | Symbol | Value |
|------|-----------|--------|-------|
| VAE Training | Num. encoder/decoder block | - | 3 |
| | Dim. hidden layer | - | 128 |
| | Num. multi-head | - | 4 |
| | Regularization weight | $\beta$ | $1e^{-6}$ |
| | Differential weight | $\lambda$ | 0.01 |
| Planner Training | Num. neighboring vehicles | $M$ | 10 |
| | Num. past timestamps | $A$ | 21 |
| | Dim. neighboring vehicles | $D_{\text{neighbor}}$ | 11 |
| | Num. lanes | - | 70 |
| | Num. points per polyline | $P$ | 20 |
| | Dim. lanes vehicles | $D_{\text{lane}}$ | 12 |
| | Num. navigation lanes | $K$ | 25 |
| | Num. encoder/decoder block | - | 3 |
| | Uncond. probability | $p$ | 0.1 |
| | Dist. weight | $\alpha$ | 0.75 |
| | Student Feat. Idx | $k$ | 3 |
| | Teacher Feat. Idx | - | 3 |
| | Dim. latent space | $L$ | 10 |
| Inference | Noise schedule | - | Linear |
| | Noise coefficient | $\beta_{\text{min}}, \beta_{\text{max}}$ | 0.1, 20.0 |
| | Temperature | - | 1.0 |
| | Guidance scale | $\omega$ | 1.0 |

a higher value indicates greater diversity. Formally, let the set of $K_{\text{mode}}$ sampled ego trajectories be $\mathcal{T}_{\text{mode}} = \{\tau^{(1)}, \tau^{(2)}, \ldots, \tau^{(K_{\text{mode}})}\}$. Each trajectory $\tau^{(i)}$ is a sequence of $T$ positions, $\tau^{(i)} = (p_i^{(1)}, p_i^{(2)}, \ldots, p_i^{(T)})$, where $p_i^{(t)}$ is the coordinate at timestep $t$. Metrics are calculated as below:

**Final Pairwise Distance (FPD).** This metric evaluates diversity at the final endpoints of the trajectories. It is the mean Euclidean distance over all $\binom{K_{\text{mode}}}{2}$ unique pairs of trajectories, calculated as:

$$\text{FPD}(\mathcal{T}_{\text{mode}}) = \frac{1}{\binom{K_{\text{mode}}}{2}} \sum_{1 \leq i < j \leq K_{\text{mode}}} \|p_i^{(T)} - p_j^{(T)}\|_2. \tag{21}$$

**Average Pairwise Distance (APD).** This metric provides a comprehensive diversity measure over the entire paths. It is the mean of the average Euclidean distances between corresponding points for all unique pairs, calculated as:

$$\text{APD}(\mathcal{T}_{\text{mode}}) = \frac{1}{\binom{K_{\text{mode}}}{2}} \sum_{1 \leq i < j \leq K_{\text{mode}}} \left( \frac{1}{T} \sum_{t=1}^{T} \|p_i^{(t)} - p_j^{(t)}\|_2 \right). \tag{22}$$

We evaluate these two metrics for both the Diffusion Planner (Zheng et al., 2025) and LAP on all scenarios of the Test14 Hard dataset, with the results presented in Table 11. As can be seen, under the same sampling method($o1s10$), LAP plans in a more compact latent semantic space, which results in its multi-modality being semantic. Consequently, it exhibits significantly more pronounced multi-modality compared to the pixel-level planner, Diffusion Planner. In addition, we further analyze the trajectory multimodality of the single-step mode. The results show that single-step decoding essentially collapses the distribution to a near "mean" trajectory, and the diversity metrics are significantly reduced compared to the multi-step setting. This confirms that the multimodal behavior of our latent planner is best expressed when using multiple denoising steps, whereas

Table 11: Comparison of planners on multimodal prediction metrics.

| Planner | APD ↑ | FPD ↑ |
|---------|-------|-------|
| Diffusion Planner($o1s10$) | 0.88 | 1.98 |
| Latent Planner($o1s10$) | **2.03** | **4.55** |
| Latent Planner($o1s1$) | 0.10 | 0.22 |

the single-step mode should be viewed as an extreme, latency-critical configuration that explicitly trades diversity for speed.

## F    EXPERIMENT METRICS DETAILS

This section provides a detailed breakdown of the official nuPlan closed-loop evaluation metrics referenced in our main experiments (Section 5). The benchmark is designed to offer a holistic assessment of a planner's real-world capabilities, encapsulating performance into a single, comprehensive score. This final score is computed for both Non-Reactive (NR) and Reactive (R) simulation scenarios. It is calculated as a weighted average of the following key performance indicators, each designed to evaluate a critical aspect of safe and efficient driving:

- *No ego at-fault collisions*: This metric evaluates the planner's ability to avoid inducing collisions for which the ego-vehicle is deemed responsible. Penalties are applied specifically to at-fault scenarios, such as collisions with stationary objects or dynamic agents where the ego-vehicle's action is the primary causal factor.

- *Time to collision within bound*: This metric quantifies the vehicle's capacity to maintain a safe temporal buffer from other road users, thereby minimizing near-miss incidents. A violation is registered if the calculated TTC, assuming constant velocity for all actors, falls below a predefined safety threshold.

- *Drivable area compliance*: This metric assesses the ego-vehicle's adherence to the designated and safe drivable surface. The planner is penalized if the vehicle's trajectory deviates from legally navigable areas, such as marked lanes and intersections.

- *Comfortableness*: This metric evaluates passenger comfort by quantifying the smoothness of the vehicle's trajectory. It penalizes undesirable kinematic behavior, specifically instances of excessive longitudinal or lateral acceleration, braking, and jerk that surpass predefined comfort thresholds.

- *Progress*: This metric measures the efficiency of the vehicle in advancing towards its navigational goal along the planned trajectory. The planner is evaluated on its ability to make sufficient progress relative to an expert-defined baseline, penalizing undue hesitation or inefficient path following.

- *Speed limit compliance*: This metric ensures the vehicle operates in accordance with posted speed regulations. A penalty is incurred if the ego-vehicle's velocity exceeds the legally mandated speed limit for its current roadway segment.

A more detailed description and calculation of the metrics can be found at `https://nuplan-devkit.readthedocs.io/en/latest/metrics_description.html`.

## G    ANALYSIS OF FEW-STEP DIFFUSION SAMPLING

In this section, we quantitatively analyze the quality of trajectories generated from few-step sampling and provide an analysis of why LAP can produce high-quality planning behaviors in just two steps.

To quantify the fidelity of few-step diffusion sampling, we compare trajectories generated using few-step processes against a 20-step reference. Recognizing that any given driving scenario can lead to multiple distinct future behaviors, we employ a dense sampling strategy. We evaluate on 10,000 diverse driving scenarios. For each scenario, we generate a large set of $B = 1000$ samples to ensure sufficient coverage of the diverse behavioral modes encoded in the latent space.

Given the multi-modal nature of the decoded trajectories, we first pair the planning behaviors that are most similar in mode before comparing their precision. Specifically, we align the sample sets by framing the task as a minimum weight bipartite matching problem. We perform this matching in the **latent space**, as it offers a more robust representation of behavioral intent than the final trajectory space. For each scenario, we generate two sets of $B = 1000$ latent vectors, $\mathcal{Z}_S = \{z_i^{(S)}\}_{i=1}^{B}$ and $\mathcal{Z}_{20} = \{z_j^{(20)}\}_{j=1}^{B}$, where $S$ denotes the denoising steps. We then employ the **Hungarian algorithm** to find the optimal permutation $\pi^*$ that minimizes the total Euclidean distance between paired latent

Table 12: Mean latent ($\mathcal{L}_z$) and trajectory ($\mathcal{L}_x$) space distances between few-step samples and a 20-step reference after mode alignment via bipartite matching. Lower values indicate higher fidelity.

| Denoising Steps (S) | Latent Distance $\mathcal{L}_z(S, 20)$ | Trajectory Distance $\mathcal{L}_x(S, 20)$ |
|---|---|---|
| $S = 1$ | 0.18 | 0.15 |
| $S = 2$ | 0.10 | 0.09 |

vectors:

$$\pi^* = \arg \min_{\pi \in \mathcal{P}_B} \sum_{i=1}^{B} \|z_i^{(S)} - z_{\pi(i)}^{(20)}\|_2,$$

where $\mathcal{P}_B$ is the set of all permutations of $\{1, 2, \ldots, B\}$.

This matching yields pairs of samples corresponding to the same behavioral mode. We then quantify the deviation using two metrics: the mean absolute distance in the **latent space** ($\mathcal{L}_z$) and the mean Euclidean distance between the corresponding **decoded trajectories** ($\mathcal{L}_x$):

$$\mathcal{L}_z(S, 20) = \frac{1}{B} \sum_{i=1}^{B} \|z_i^{(S)} - z_{\pi^*(i)}^{(20)}\|_1,$$

$$\mathcal{L}_x(S, 20) = \frac{1}{B} \sum_{i=1}^{B} \left( \frac{1}{T_{eval}} \sum_{t=1}^{T_{eval}} \|x_i^{(S)}(t) - x_{\pi^*(i)}^{(20)}(t)\|_2 \right)$$

$\mathcal{L}_x$ is computed over the first $T_{eval} = 10$ timesteps (1 second), as the accuracy of the near-term path is most critical for immediate planning decisions and ensuring reactive safety. The results are summarized in Table 12. We observe that while single-step decoding yields latent representations with a 0.18 average per-dimension discrepancy from the reference, the smoothness of the latent space ensures the decoded trajectories achieve a low average displacement deviation of just $0.15m$. This level of error has a negligible impact on the immediate decision-making process. Subsequently, employing a two-step decoding process further reduces this error, leading to enhanced model performance.

## H  FINE-GRAINED FEATURE DISTILLATION MODULE DETAILS

In this section, we present the implementation details of fine-grained feature distillation module. As shown in Fig. 10, for each training sample, we feed the ground-truth trajectory $\mathcal{T}$ and scene condition $C$ into the pre-trained pixel-level Diffusion Planner (Zheng et al., 2025) $f$ and take the feature of its intermediate DiT block as the target feature

$$\boldsymbol{y}^* = f(\mathcal{T}, \boldsymbol{C}) \in \mathbb{R}^{(1+M) \times D},$$

which encodes fine-grained trajectory–scene interactions in the original vectorized space. In parallel, we sample a noisy latent $\boldsymbol{z}_t$ from the forward diffusion process of the corresponding latent trajectory, run our latent DiT planner conditioned on $\boldsymbol{C}$, and extract an intermediate representation

$$\boldsymbol{h}_k \in \mathbb{R}^{(1+M) \times D}$$

from the $k$-th DiT block. A lightweight MLP projection head $h_\phi$ is applied on top of $\boldsymbol{h}_k$ to map the student features into the same feature space as $\boldsymbol{y}^*$, and we minimize a mean-squared distillation loss:

$$\mathcal{L}_{\text{dist}} = \mathbb{E}_{\mathcal{T}, \, \boldsymbol{z}_0 \sim q(\boldsymbol{z}|\mathcal{T}), \, t \sim \mathcal{U}(0,1), \, \boldsymbol{z}_t \sim q_t(\boldsymbol{z}_t|\boldsymbol{z}_0)} \left[ \|h_\phi(\boldsymbol{h}_k) - \boldsymbol{y}^*\|^2 \right], \tag{23}$$

which is combined with the standard diffusion loss as

$$\mathcal{L} = \mathcal{L}_{\text{diff}} + \alpha \mathcal{L}_{\text{dist}}, \tag{24}$$

where $\alpha > 0$ controls the strength of distillation. During training, the teacher is kept fixed and only provides the target features $\boldsymbol{y}^*$; at inference time, the teacher is discarded and only the latent planner is used, so the distillation introduces no additional test-time cost while encouraging the

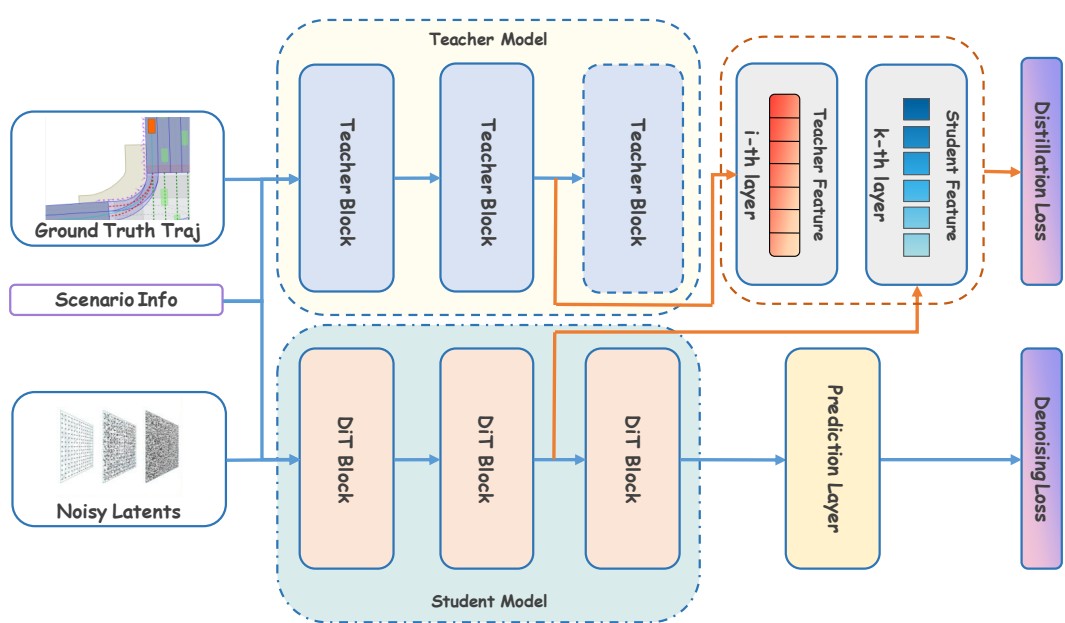

Figure 10: Fine-grained Feature Distillation Module.

student's intermediate representations to inherit the teacher's fine-grained, pixel-level understanding of trajectory–scene interactions.

To verify the importance of having a feature representation that properly encodes trajectory–scene interaction information, we conduct an additional experiment: we directly use the raw trajectory $\mathcal{T}$ as a pixel-level supervision signal and replace the distillation loss with:

$$\mathcal{L}_{\text{pixel}} = \mathbb{E}_{\mathcal{T},\, \boldsymbol{z}_0 \sim q(\boldsymbol{z}|\mathcal{T}),\, t \sim \mathcal{U}(0,1),\, \boldsymbol{z}_t \sim q_t(\boldsymbol{z}_t|\boldsymbol{z}_0)} \left[ \|h_{\text{pred}}(\boldsymbol{h}_k) - \mathcal{T}\|^2 \right] \quad (25)$$

where $h_{\text{pred}}$ is the prediction head in the pixel space (the same as in Diffusion Planner). This is equivalent to directly using the raw, unencoded trajectories as the regularization target. The experimental results are reported in Table 13. We observe that directly using the raw trajectories as an auxiliary decoding target actually degrades performance, indicating that naively imposing a pixel-level planning loss on the latent space is harmful. This highlights that **a feature representation encoding fine-grained trajectory–scene interaction information is crucial**.

Table 13: Impact of Different Distillation Targets.

| Distillation Target | Test14-Hard(NR) | Test14-Hard(R) |
|---|---|---|
| - | 74.13 | 66.31 |
| Raw Traj. | 74.62 | 63.5 |
| Encoded Feat. | 76.49 | 68.89 |

## I  TRAINING STABILITY

Our training pipeline follows a two-stage procedure: we first train the trajectory VAE with a reconstruction + KL + differential loss, and then train the conditional diffusion model on the resulting latent codes. This decoupling makes optimization straightforward and empirically stable; with our default hyperparameters, both stages converge smoothly without divergence or mode collapse. In particular, we normalize trajectories and rescale the target latents by the inverse of their global standard deviation, a technique previously introduced to stabilize optimization in the latent space (Rombach et al., 2022). We further provide the training loss curve(Fig. 11) and test score curve(Fig. 12),

from which we observe that the training loss decreases smoothly, while the test score increases with mild fluctuations and peaks at around 140 epochs.

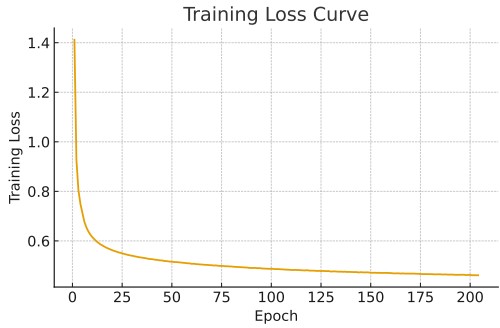

Figure 11: Training Loss vs. Epoch

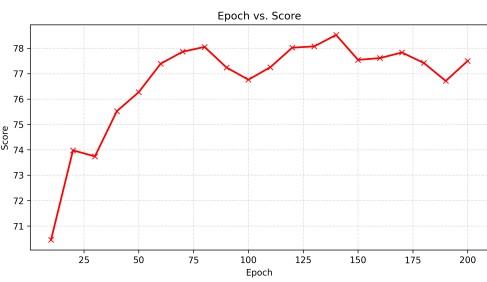

Figure 12: Test Score vs. Epoch

## J   LIMITATIONS & FUTURE WORK

In this section, we present the limitations of this work and potential directions for future work.

**Dependence on Latent Space Structure.** The overall performance of LAP is heavily contingent on the quality of the latent space learned by the Variational Autoencoder. A suboptimal latent space could constrain the planner's ability to capture the full diversity of complex driving strategies or result in decreased fidelity of the reconstructed trajectories.

*Future Work:* Planning in latent space is sensitive to the latent space structure. Therefore, a possible solution is to jointly optimize the latent space structure and the latent diffusion model in an end-to-end manner using REPA-E (Leng et al., 2025).

**Reliance on a Pixel-Level Diffusion Teacher.** Our model currently relies on a pixel-level diffusion model as a feature extractor to guide the interaction between the planning space and conditional information, which consumes significant additional computational resources.

*Future Work:* A possible direction for future work is to introduce additional modules or fine-grained auxiliary tasks to guide a more fine-grained interaction between the model and the input conditions.

