# OpenReview forum: "LAP: Fast $\textbf{LA}$tent Diffusion $\textbf{P}$lanner with Fine-Grained Feature Distillation for Autonomous Driving"
_ICLR.cc/2026/Conference — Submitted to ICLR 2026_

### Official Review · Reviewer_MsQH · 2025-10-30

**Soundness:** 3
**Presentation:** 3
**Contribution:** 3
**Rating:** 6
**Confidence:** 4

**Summary:**

This paper presents LAP (LAtent Planner), a novel planning framework for autonomous driving. The core idea is to first use a Variational Autoencoder (VAE) to learn a disentangled latent space that separates high-level strategic intents from low-level kinematic details. A latent diffusion model is then trained in this compact space to generate plans. The authors also introduce a "fine-grained feature distillation" mechanism to enhance the interaction between the planner's semantic representation and the vectorized scene context. The method is shown to achieve state-of-the-art (SOTA) closed-loop performance on the nuPlan benchmark while being highly efficient, capable of generating plans in a single denoising step.

**Strengths:**

1. The paper introduces a novel and elegant architecture. Applying diffusion models to a learned latent space for planning is a promising direction. The conceptual decoupling of high-level strategy and low-level control is well-motivated and addresses a key challenge in end-to-end planning.

2. The claimed SOTA performance on a large-scale, closed-loop benchmark like nuPlan is impressive. This provides strong validation for the proposed framework and its practical effectiveness.

3. A major contribution is the model's ability to operate in a single denoising step. This directly tackles the primary drawback of diffusion models (high latency) and makes the approach far more viable for real-time applications in autonomous driving.

**Weaknesses:**

1. The "fine-grained feature distillation" is a key component, but its exact architectural details are slightly underdeveloped in the main text. A more detailed diagram or explanation in the appendix would be beneficial.

2. The paper's premise relies on the VAE's ability to achieve disentanglement. While the strong downstream results imply this is successful, the paper would be more compelling with a brief qualitative analysis (e.g., latent space interpolation) to visually demonstrate this property.

3. The relationship between the "single-step" inference and the "multi-modal" nature of the planner could be made more explicit. A brief discussion on how plan diversity is (or is not) preserved in this fast-inference mode would be helpful.

**Questions:**

1. Could you please elaborate on the single-step inference mode? How does it maintain the ability to generate diverse, multi-modal plans, which is a key benefit of diffusion models? Or does it produce a single, high-quality "mean" trajectory?

2. Can you provide more implementation details on the feature distillation module? What design choices were most critical to its success?

---

> ### Author Response · Authors · 2025-11-20
>
> We thank the reviewer for the constructive comments. Regarding the concerns of the reviewer MsQH, we provide the following responses.
>
> > **W1 & Q2. Implementation details on the feature distillation module. What design choices were most critical to its success?**
>
> (1) We thank the reviewer for pointing out that the description of the fine-grained feature distillation module is currently under-specified. In the [revised version](https://openreview.net/pdf?id=uHEaVkj8I3), we have added a detailed diagram (Fig. 10) and explanation in Appendix H, which we briefly outline here:
>
> - For each training sample, we feed the ground-truth trajectory $ T $ and scene condition $ C $ into a pre-trained pixel-space Diffusion Planner (teacher) and extract its intermediate DiT-block feature $ y^{*} = f(T, C) $.
>
> - In parallel, we run our latent-space DiT planner (student) on the corresponding noisy latent $ \boldsymbol{z}_t $ and condition $ C $, and take an intermediate representation $ h_k $ from the $ k $-th DiT block.
>
> - We apply a small MLP head $ h_\phi $ on top of $ h_k $ to map the student features into the same space as the teacher feature $ y^{*} $.
>
> - We compute a feature-level distillation loss $L_{\mathrm{dist}} = \mathbb{E}\left[ || h_{\phi}(h_k) - y^{*} ||^2 \right]$, and combine it with the standard diffusion loss as $L = L_{\mathrm{diff}} + \alpha L_{\mathrm{dist}}$.
>
>
> - During training, the teacher is kept frozen and only provides $y^{\*}$; during inference, the teacher is discarded and we only use the latent planner, so the distillation does not introduce extra test-time cost.
>
> (2) Its success critically hinges on providing **a feature representation that well encodes fine-grained trajectory–scene interaction information**. To further validate this, we conduct a small ablation in which we replace the distillation target with the raw trajectories as a regularization term. As shown in the table below, this modification actually leads to degraded performance, indicating that naively adding a pixel-level regularization term is ineffective, and that a well-encoded feature representation is the key.
>
> | Distillation Traget                  | Test14-Hard(NR) | Test14-Hard(R) |
> |:--------------------------|:----------------------:|:--------:|
> | -        | 74.13  | 66.31   |
> | Raw Traj.        | 74.62  | 63.5   |
> | Traj-Scene Feat.    | 76.49  | 68.89  |
>
> The relevant details and results have been added to Appendix H in the [revised version](https://openreview.net/pdf?id=uHEaVkj8I3).
>
> > **W2. The paper would be more compelling with a brief qualitative analysis to visually demonstrate the VAE's ability to achieve disentanglement.**
>
> We thank the reviewer for the suggestion to add a figure illustrating the learned latent space.
>
> - The current version already includes a latent-space interpolation visualization in Appendix C, which qualitatively shows smooth transitions between trajectories.
>
> - To better demonstrate the properties of the VAE latent space, we have additionally conducted two experiments:
>   1. **Clustering in the latent space:** We perform clustering directly on the latent vectors and decode each cluster centroid back to the trajectory space, obtaining the results shown in Fig. 6. One can observe that trajectories within the same latent cluster share similar motion patterns, indicating that the latent space captures meaningful structure.
>   2. **Clustering in the original trajectory space + latent UMAP:** We cluster trajectories in the original trajectory space(Fig. 7) and treat the cluster IDs as “intention” labels. We then visualize the corresponding latent vectors using UMAP[1], colored by these labels, as shown in Fig. 8. The resulting plot exhibits well-separated clusters, which indicates that the latent space successfully capture the intention categories. We also conduct a further visual analysis of the outliers(Fig. 9), confirming that they correspond to the special intent of “staying still”.
>
> These results together suggest that the VAE has learned a latent space with good smoothness and semantic disentanglement. All details and results of the visualization experiments have been added to Appendix C in the [revised version](https://openreview.net/pdf?id=uHEaVkj8I3).

---

> ### Author Response · Authors · 2025-11-20
>
> > **W3 & Q1. The relationship between the "single-step" inference and the "multi-modal" nature of the planner could be made more explicit. A brief discussion on how plan diversity is (or is not) preserved in this fast-inference mode would be helpful.**
>
> We thank the reviewer for the insightful question. We have further analyzed the behavior of the single-step inference mode in terms of plan diversity. The quantitative multi-modality results are summarized in the table below, where APD and FPD denote the Average Pairwise Distance and Final Pairwise Distance, respectively. The results show that **single-step decoding essentially collapses the distribution to a near “mean” trajectory**, the diversity metrics are significantly reduced compared to the multi-step setting. This confirms that the multi-modal nature of our diffusion planner is best expressed when using multiple denoising steps, whereas the single-step mode should be viewed as an extreme, latency-critical configuration that trades off diversity for speed. We have added this result to Appendix E in the [revised version](https://openreview.net/pdf?id=uHEaVkj8I3), which also includes additional quantitative analyses of multi-modality as well as further experimental details.
>
>
> | Planner                  | APD | FPD |
> |:--------------------------|:----------------------:|:--------:|
> | Latent Planner(o1s10)    | 2.03  | 4.55  |
> | Latent Planner(o1s1)    | 0.10  | 0.22  |
>
> [1] Leland McInnes, John Healy, and James Melville. Umap: Uniform manifold approximation and
> projection for dimension reduction. arXiv preprint arXiv:1802.03426, 2018.

---

> ### Author Response · Authors · 2025-11-28
> **Reminder for rebuttal**
>
> Dear Reviewer MsQH,
>
> Only six days remain before the discussion phase ends. We wanted to check whether we have addressed all of your concerns about the paper. We would greatly welcome any additional feedback or suggestions you may have.
>
> Best regards,
>
> Authors of Submission 8441

---

### Official Review · Reviewer_iCPG · 2025-10-30

**Soundness:** 3
**Presentation:** 3
**Contribution:** 2
**Rating:** 4
**Confidence:** 4

**Summary:**

The paper proposes LAtent Planner (LAP), a latent diffusion framework for motion planning in autonomous driving. It first trains a Trajectory VAE to learn a compact latent space disentangling high-level semantics from low-level kinematics, and then performs conditional diffusion in this latent space. To bridge the gap between the latent semantic space and the vectorized perception features, the authors introduce a fine-grained feature distillation module, using features from a teacher diffusion planner as guidance. LAP claims to significantly improve both planning quality and inference efficiency, achieving state-of-the-art closed-loop scores on the nuPlan benchmark with up to 10× faster inference than existing diffusion-based planners.

**Strengths:**

- Motivation clarity: The paper articulates a clear motivation — diffusion planners suffer from high latency and focus too much on low-level trajectory details. Planning in latent space is a logical response.
- Efficiency gains: The reported inference speedup (10× faster) is notable and relevant for real-time deployment in driving systems.

**Weaknesses:**

- The paper claims that operating in latent space improves semantic modeling, but does not provide meaningful analysis of the learned latent representations (e.g., clustering by intent, diversity metrics). The VAE–diffusion interface is treated as a black box.
- Closed-loop performance improvements over Diffusion Planner are marginal (≤1–2%), which may fall within the noise margin of nuPlan’s stochastic simulator.

**Questions:**

- How stable is the VAE–diffusion training pipeline? Does the VAE reconstruction error correlate with downstream planning performance?
- How does the model handle OOD (out-of-distribution) scenarios where latent-space priors may fail to represent unseen semantics?
- Can the authors provide evidence that the latent space truly captures strategic semantics (e.g., intent categories) rather than just being a compressed waypoint representation?

---

> ### Author Response · Authors · 2025-11-20
>
> We thank the reviewer for the constructive comments. Regarding the concerns of the reviewer iCPG, we provide the following responses.
>
> > **W1 & Q3. The paper claims that operating in latent space improves semantic modeling, but does not provide meaningful analysis of the learned latent representations.**
>
> We thank the reviewer for raising this point. In the [revised version](https://openreview.net/pdf?id=uHEaVkj8I3), we have added more visual analyses of the latent space in Appendix C：
>
> - First, we perform **k-means clustering on the ego-trajectory latent codes**(k = 10) and decode each cluster centroid back to the trajectory space. As shown in Fig. 6, the decoded prototype trajectories correspond to clearly distinct high-level maneuvers (e.g., going straight with different velocities, strong braking, and left/right turning behaviors). This indicates that trajectories are organized in the latent space according to strategic driving intent rather than low-level waypoint coordinates.
>
> - Second, we cluster trajectories directly in the original trajectory space(Fig. 7) and use the resulting cluster index as a coarse “trajectory intent” label for each sample. We then **project the corresponding latent vectors to 2D using UMAP[1] and color each point by its intent label** (Fig. 8). We observe that trajectories sharing the same intent form compact and mostly non-overlapping regions in the latent space, while different intents occupy well-separated areas. This demonstrates that the learned latent codes are highly correlated with high-level semantic modes of driving. We also conduct a further **visual analysis of the outliers (Fig. 9), confirming that they correspond to the special intent of “staying still.”**
>
> Together with the latent interpolation experiment already included in the paper(Fig. 5), which shows smooth and plausible transitions between different trajectories when interpolating in latent space, these new analyses provide direct evidence that our VAE learns a compact, smooth, and semantically meaningful latent representation. This supports our claim that operating in the latent space improves semantic modeling for planning.
>
> > **W2. Closed-loop performance improvements over Diffusion Planner are marginal (≤1–2\%), which may fall within the noise margin of nuPlan’s stochastic simulator.**
>
> We thank the reviewer for raising this concern. We would like to clarify that, under our evaluation setup, the variability mainly comes from the stochastic sampling of diffusion-based planners rather than from the nuPlan simulator itself.
>
> - **Determinism of the simulator.** We re-ran the official nuPlan closed-loop evaluation for a non-diffusion planner (UrbanDriver[2]) three times with different random seeds(0, 3407, 3607) and obtained identical scores(43.89) across all runs. This indicates that the simulator and evaluation pipeline are effectively deterministic for a fixed planner implementation, and that any run-to-run variation in our experiments stems from the diffusion sampling procedure inside the planner rather than from simulator noise.
>
> - **Evaluation protocol of LAP.** The LAP results reported in Table 1 are already **averaged over 5 independent closed-loop runs**. We report the corresponding minimum and maximum values in the table below. We acknowledge that on the Val14 split LAP is slightly worse or comparable to Diffusion Planner (e.g., 89.37 vs. 89.64 on Val14(NR), and 82.23 vs. 82.80 on Val14(R)). However, on the more challenging Test14-hard benchmarks LAP exhibits clear and stable gains. On the Non-Reactive Test14-hard benchmark, our 5 runs range from 77.83 to 78.89, with the reported score 78.52 being their mean. In contrast, on the same benchmark Diffusion Planner achieves 75.44, so LAP improves the Non-Reactive Test14-hard score from 75.44 → 78.52, i.e. +3.08 absolute points (≈4.1\% relative). Even in the worst LAP run (77.83), the score remains about 2 points higher than Diffusion Planner (75.44), which is well beyond the run-to-run variation we observe from diffusion sampling alone. The gains are consistently larger than the fluctuations we observe due to diffusion sampling, and are therefore unlikely to be explained by simulator noise alone. We hypothesize that planning in the latent representation space helps LAP handle complex and challenging scenarios more robustly, which is better reflected on Test14-hard than on Val14.
>
> | Data Type | Val14 (NR) | Val14 (R) | Test14-hard (NR) | Test14-hard (R) | Test14 (NR) | Test14 (R) |
> |:-----------|:-----------:|:-----------:|:-----------:|:-----------:|:-----------:|:-----------:|
> | Average   | 89.37      | 82.23     | 78.52            | 70.49           | 90.42       | 85.12      |
> | Max       | 89.51      | 82.42     | 78.89            | 70.97           | 90.83       | 85.57      |
> | Min       | 88.97      | 82.08     | 77.83            | 70.05           | 89.75       | 84.66      |

---

> ### Author Response · Authors · 2025-11-20
>
> > **Q1. How stable is the VAE–diffusion training pipeline? Does the VAE reconstruction error correlate with downstream planning performance?**
>
> - **Training Stability**: Our training pipeline is a two-stage procedure: we first train the trajectory VAE with a reconstruction + KL + differential loss, and then train the conditional diffusion model on the resulting latent codes. This decoupling makes optimization straightforward and empirically stable. In particular, we normalize trajectories and rescale the target latents by the inverse of their global standard deviation, which was specifically introduced to stabilize optimization in the latent space[3]. We have **added training/test curves(Fig. 11/12) for the VAE-Diffusion Training in Appendix I** in the [revised version](https://openreview.net/pdf?id=uHEaVkj8I3). We observe that the training loss decreases smoothly, while the test score increases with mild fluctuations and peaks at around 140 epochs. To further verify the stability of training, we **reran the entire training pipeline with three additional random seeds (41, 42, 43) under the same settings**. Across these runs, **the final performance varies by less than 1\% on all splits** (Val14 NR: 89.40±0.11, Val14 R: 82.17±0.18, Test14-hard NR: 78.54±0.32, Test14-hard R: 70.35±0.44, Test14-random NR: 90.56±0.27, Test14-random R: 85.17±0.31), indicating that the pipeline is stable.
>
> - **Correlation between VAE reconstruction error and downstream planning performance**: we find that good reconstruction is **necessary but not sufficient**. The VAE effectively sets an upper bound on planning quality: if it cannot reconstruct kinematically feasible trajectories, the planner will inevitably fail. However, reconstruction error does not correlate monotonically with closed-loop performance. As shown below, a VAE with latent dimension $L=10$ trained for 200 epochs (“L10Epo200”) yields better reconstruction (Table 9 in Appendix C.3) than the 120-epoch counterpart (“L10Epo120”), yet the corresponding planner performs worse in closed-loop planning (Test14-hard NR: 76.38 → 74.72, Test14-hard R: 68.28 → 66.69). This suggests that **aggressively minimizing reconstruction error causes the VAE to overfit training trajectories, leading to a less semantically structured latent space**. In contrast, the early-stopped VAE (120 epochs) yields a more compact latent representation, which in turn leads to better downstream planning.
>
> | VAE         | Reconstruction Error(m) | Test14-Hard(NR) | Test14-Hard(R) |
> |:--------------------------|:---------------------:|:--------:|:--------:|
> | L10Epo200    | 0.170 | 74.72  | 66.69   |
> | L10Epo120    | 0.219 | 76.38  | 68.28  |
>
> > **Q2. How does the model handle OOD scenarios where latent-space priors may fail to represent unseen semantics?**
>
> We address out-of-distribution (OOD) scenarios at both the **data** and **representation** levels.
>
> - **Data augmentation targeted at OOD generalization.** Following previous work[4], we adopt the same trajectory-level data augmentation strategy when training the VAE and the latent diffusion model. This augmentation was explicitly designed in previous work to improve robustness to distribution shift in nuPlan’s closed-loop evaluations, and we inherit these augmentations to expose LAP to a wider variety of behaviors and scene layouts than what appears in the raw logs.
>
> - **Early-stopped, compact latent space for better generalization.** As discussed in Q1, we deliberately early stop VAE training at 120 epochs instead of using the lowest reconstruction error checkpoint at 200 epochs. Although the 200-epoch VAE reconstructs trajectories more accurately (Table 9), using it as the latent prior hurts closed-loop performance (Table 8). This indicates that the more heavily trained VAE tends to overfit the training distribution, creating a latent space that is less robust when encountering OOD semantics in closed-loop simulation. In contrast, the early-stopped VAE empirically generalizes better to unseen combinations of traffic participants and scene geometry.
>
> [1] Leland McInnes, John Healy, and James Melville. Umap: Uniform manifold approximation and
> projection for dimension reduction. arXiv preprint arXiv:1802.03426, 2018.
>
> [2] Oliver Scheel, Luca Bergamini, Maciej Wołczyk, Bła˙zej Osi´nski, and Peter Ondruska. Urban driver:
> Learning to drive from real-world demonstrations using policy gradients, 2021.
>
> [3] Robin Rombach, Andreas Blattmann, Dominik Lorenz, Patrick Esser, and Bj¨orn Ommer. High-
> resolution image synthesis with latent diffusion models. In Proceedings of the IEEE/CVF confer-
> ence on computer vision and pattern recognition, pp. 10684–10695, 2022.
>
> [4] Yinan Zheng, Ruiming Liang, Kexin Zheng, Jinliang Zheng, Liyuan Mao, Jianxiong Li, Weihao
> Gu, Rui Ai, Shengbo Eben Li, Xianyuan Zhan, et al. Diffusion-based planning for autonomous
> driving with flexible guidance. arXiv preprint arXiv:2501.15564, 2025.

---

> ### Author Response · Authors · 2025-11-28
> **Reminder for rebuttal**
>
> Dear Reviewer iCPG,
>
> Only six days remain before the discussion phase ends. We wanted to check whether we have addressed all of your concerns about the paper. We would greatly welcome any additional feedback or suggestions you may have.
>
> Best regards,
>
> Authors of Submission 8441

---

### Official Review · Reviewer_dMwV · 2025-10-31

**Soundness:** 3
**Presentation:** 2
**Contribution:** 2
**Rating:** 4
**Confidence:** 4

**Summary:**

This paper propose to learn planning task in latent space, disentangles  high-level intents from low-level kinematics, to capture rich, multi-modal driving strategies. By designing a two stage diffusion-based planner, this paper achieves good performance and inference speed-up.

**Strengths:**

* A trajectory VAE is proposed to learn a semantic latent space which disentangles high-level strategic semantics from low-level kinematic execution.
* A latent diffusion model is designed to learn planning task in high-level semantic latent space, and a feature distillation method is proposed to bridge the gap between semantic space and vectorized scene perception.
* The whole framework, thanks to the two-stage design, achieves 10x speed-up than baseline.

**Weaknesses:**

* Though the paper claims the planning should be learned in a high-level semantic, the reason is not well described in the paper(only the visualization result in Fig.3 seems not enough)， making the motivation not clear enough.
* The inference speed is fast, but more details are needed in the paper, e.g. model size, FLOPS.
* First replace low-level planning space with high-level semantic space, but then introduce another feature distillation module to align these two spaces, makes the framework complicated and seems unnecessary.

**Questions:**

See Weaknesses.

---

> ### Author Response · Authors · 2025-11-20
>
> We thank the reviewer for the constructive comments. Regarding the concerns of the reviewer dMwV, we provide the following responses.
>
> > **W1. Though the paper claims the planning should be learned in a high-level semantic, the reason is not well described in the paper，making the motivation not clear enough.**
>
> We thank the reviewer for highlighting that the motivation for learning planning in a high-level semantic space is not sufficiently emphasized. Our motivation for learning planning in a high-level semantic latent space is closely aligned with three practical requirements for autonomous driving planners: (i) accurately modeling multi-modal strategic behavior, (ii) achieving real-time inference, and (iii) maintaining robustness in challenging scenarios.
>
> - First, raw waypoint space is dominated by redundant kinematic constraints, while the true uncertainty lies in a much lower-dimensional space of strategic intents. By compressing trajectories into a semantic latent space and letting the decoder handle low-level kinematics, the diffusion model can focus on this intent manifold, which leads to more structured and diverse multi-modal plans, as quantified in Fig. 3.
>
> - Second, diffusion in this compact latent space significantly reduces the number of function evaluations required, enabling substantially faster inference that is compatible with real-time deployment as shown in Table 2.
>
> - Third, a semantically organized latent space allows the planner to better capture rare but safety-critical behaviors, resulting in higher closed-loop performance particularly on challenging scenarios, as shown in Table 1 (Test14-hard NR/R).
>
> In the [revised version](https://openreview.net/pdf?id=uHEaVkj8I3), to better highlight the motivation behind our approach, we have added a dedicated subsection titled “Latent Planning vs. Pixel-level Planning” in Section 5.2. This subsection provides an intuitive side-by-side comparison in tabular form, and the table is shown below:
>
> | Planner                  | Test14-hard(NR) | Inference time(ms) | APD | FPD |
> |:--------------------------|:----------------------:|:--------:|:--------:| :--------:|
> | Diffusion Planner        | 75.44  | 202.60   | 0.88 | 1.98 |
> | Latent Planner           | 78.52  | 21.69    | 2.03 | 4.55 |
>
> > **W2. More details are needed in the paper, e.g. model size, FLOPS.**
>
> We thank the reviewer for the suggestion to make the paper more comprehensive. In the [revised version](https://openreview.net/pdf?id=uHEaVkj8I3), we have added the model size and GFLOPs of the state-of-the-art methods, which are summarized in the following table:
>
> | Planner                  | Inference time (ms) ↓ | Score ↑ | GFLOPs | Model size (M) |
> |:-------------------------|:---------------------:|:-------:|:------:|:--------------:|
> | UrbanDriver[1]           | 67.47                 | 43.89   |   -    |       -        |
> | GC-PGP[2]                | 39.24                 | 46.13   |   -    |       -        |
> | Pluto w/o refine[3]      | 745.87                | 73.61   | 0.994  |      4.24      |
> | DiffusionPlanner[4]      | 202.60                | 75.44   | 1.38   |      6.04      |
> | Latent Planner (o1s1)    | 18.81                 | 78.11   | 1.30   |      7.03      |
> | Latent Planner (o1s2)    | 21.69                 | 78.52   | 1.33   |      7.03      |
>
> As shown in the table, although LAP has a similar GFLOPs count to Diffusion Planner (most GFLOPs are spent in the scene encoder), it requires far fewer serial decoding steps. This allows most computations to run in parallel on the GPU, and therefore LAP enjoys a clear advantage in inference speed.

---

> ### Author Response · Authors · 2025-11-20
>
> > **W3. First replace low-level planning space with high-level semantic space, but then introduce another feature distillation module to align these two spaces, makes the framework complicated and seems unnecessary.**
>
> We thank the reviewer for raising this concern, and we hope to clarify the motivation and role of the feature distillation module as follows：
>
> - Our framework is **centered on performing planning in a high-level semantic space**, which allows the model to better capture the underlying multimodal driving strategies. The strong performance on the challenging Test14-hard benchmark demonstrates that planning in this high-level semantic space improves decision-making in complex scenarios.
>
> - All other components, including the feature distillation module, are **designed specifically to support planning in this high-level space rather than to re-introduce low-level planning**. The latent space abstracts away detailed kinematics, which improves efficiency and makes it easier to model multimodal strategies, but it also makes it harder for the planner to stay geometrically aligned with the vectorized scene representation. The distillation term is specifically designed to serve as a bridge between these two modalities, rather than as a simple alignment.
>
> - Our ablation studies show that **using features from an early layer of the teacher (the first layer in our experiments) as the distillation target can also significantly improve the student’s performance (Table 7), even though the teacher itself does not rely on this layer for its final prediction**. Moreover, **enforcing alignment at other layer of the student network—not only at the final decision layer—consistently improves performance (Table 6)**. These results indicate that **the distillation module improves final performance primarily by guiding the student to learn better intermediate-layer representations**, which we interpret as promoting interaction between the high-level planning space and the low-level scene space. This is consistent with recent findings in the image generation literature[5].
>
> - Importantly, the feature distillation module is **used only as an auxiliary loss during training and does not change the inference-time architecture or complexity**. The ablation in Table 4 shows that adding the distillation term on top of initial state injection improves the Test14-hard closed-loop scores from 74.13→76.49 (NR) and 66.31→68.89 (R), and Table 6 further demonstrates that these gains are consistent across different distillation settings.
>
> - To further demonstrate that simply reintroducing low-level planning is ineffective, we add an experiment where the distillation term is replaced with a pixel-level trajectory prediction term, which serves as a pixel-level regularizer. The results are shown in the table below. We observe that this naïve reintroduction of low-level planning actually degrades model performance, indicating that teacher features that properly encode trajectory–scene interactions are essential for effective planning in the latent space. We have added the experimental details in Appendix H in the [revised version](https://openreview.net/pdf?id=uHEaVkj8I3).
>
> | Distillation Target    | Test14-Hard(NR) | Test14-Hard(R) |
> |:--------------------------|:----------------------:|:--------:|
> | -        | 74.13  | 66.31   |
> | Raw Traj.        | 74.62  | 63.5   |
> | Traj-Scene Feat.    | 76.49  | 68.89  |
>
> In the [revised version](https://openreview.net/pdf?id=uHEaVkj8I3), we have added a short subsection, “DISCUSSION ON FEATURE DISTILLATION MODULE”, in Sec. 5.4 to analyze and discuss the actual role of the feature distillation module.
>
> [1] Oliver Scheel, Luca Bergamini, Maciej Wołczyk, Bła˙zej Osi´nski, and Peter Ondruska. Urban driver:
> Learning to drive from real-world demonstrations using policy gradients, 2021.
>
> [2] Marcel Hallgarten, Martin Stoll, and Andreas Zell. From prediction to planning with goal condi-
> tioned lane graph traversals. In 2023 IEEE 26th International Conference on Intelligent Trans-
> portation Systems (ITSC), pp. 951–958. IEEE, 2023.
>
> [3] Jie Cheng, Yingbing Chen, and Qifeng Chen. Pluto: Pushing the limit of imitation learning-based
> planning for autonomous driving. arXiv preprint arXiv:2404.14327, 2024.
>
> [4] Yinan Zheng, Ruiming Liang, Kexin Zheng, Jinliang Zheng, Liyuan Mao, Jianxiong Li, Weihao
> Gu, Rui Ai, Shengbo Eben Li, Xianyuan Zhan, et al. Diffusion-based planning for autonomous
> driving with flexible guidance. arXiv preprint arXiv:2501.15564, 2025.
>
> [5] ihyun Yu, Sangkyung Kwak, Huiwon Jang, Jongheon Jeong, Jonathan Huang, Jinwoo Shin, and
> Saining Xie. Representation alignment for generation: Training diffusion transformers is easier
> than you think. arXiv preprint arXiv:2410.06940, 2024.

---

> ### Author Response · Authors · 2025-11-28
> **Reminder for rebuttal**
>
> Dear Reviewer dMwV,
>
> Only six days remain before the discussion phase ends. We wanted to check whether we have addressed all of your concerns about the paper. We would greatly welcome any additional feedback or suggestions you may have.
>
> Best regards,
>
> Authors of Submission 8441

---

### Author Response · Authors · 2025-11-20

Dear AC and reviewers,

We thank all reviewers for their valuable time and insightful feedback. Following your suggestions, we have added new experiments, analyses, and explanations to better demonstrate the advantages of LAP from multiple perspectives. Please refer to our [updated paper](https://openreview.net/pdf?id=uHEaVkj8I3), where all changes are highlighted in blue. We will also open-source our code so that the community can further explore trajectory planning in the latent space.

We hope our responses address your concerns. If you have any further questions or would like additional clarifications, we are more than happy to provide them and continue improving the paper.

Best regards,
Authors of Submission 8441

---

### Author Response · Authors · 2025-11-26

Dear Reviewers,

As the deadline of author-reviewer discussion is approaching, If you have any further concerns or questions to discuss, we are more than willing to address them.

Thanks,

Authors of Submission 8441

---

### Author Response · Authors · 2025-12-01

# General Response

We thank all reviewers for their thoughtful and constructive feedback.

We are glad that the strengths of our work were recognized, including:

- *“A trajectory VAE … learn a semantic latent space, disentangling high-level semantics from low-level kinematics.”* — **R-dMwV**
- *“A latent diffusion model … and a feature distillation method … bridge the gap between semantic space and vectorized scene perception.”* — **R-dMwV**
- *“Planning in latent space is a logical response”* and the *“10× faster”* inference is *“notable and relevant for real-time deployment.”* — **R-iCPG**
- *“A novel and elegant architecture …” and "The conceptual decoupling of high-level strategy and low-level control is well-motivated and addresses a key challenge in end-to-end planning"* — **R-MsQH**
- *“SOTA performance on … nuPlan is impressive, providing strong validation for the proposed framework and its practical effectiveness”* and the ability to operate in *“a single denoising step”* make the approach *“viable for real-time applications.”* — **R-MsQH**

---

# Updates and New Experiments

Below we summarise the main changes and additions made in the revised version.

### 1. Motivation & latent vs. pixel-level planning

We added a short subsection **“Latent Planning vs. Pixel-level Planning”** (Sec. 5.2) and a direct comparison to Diffusion Planner, showing that planning in latent space improves robustness and multi-modality while being much faster on Test14-hard:

**Table 1. Latent vs. pixel-level planning (Test14-hard).**

| Planner           | Score (NR) ↑ | Time (ms) ↓ | APD ↑ | FPD ↑ |
|:------------------|:------------:|:-----------:|:-----:|:-----:|
| Diffusion Planner | 75.44        | 202.60      | 0.88  | 1.98  |
| Latent Planner    | **78.52**    | **21.69**   | **2.03** | **4.55** |

We also report GFLOPs and model size for all methods, showing that LAP has similar GFLOPs to Diffusion Planner but much lower latency due to fewer serial denoising steps.

### 2. Feature distillation module

We added a detailed diagram and description in Appendix H, and new ablations showing that feature distillation is a **training-only auxiliary loss** that improves trajectory–scene interaction modeling:

**Table 2. Effect of different distillation targets (Test14-hard).**

| Distillation Target | NR ↑ | R ↑  |
|:--------------------|:----:|:----:|
| None                | 74.13| 66.31|
| Raw Traj.           | 74.62| 63.50|
| Traj–Scene Feat.    | **76.49** | **68.89** |

This confirms that distilling rich teacher features is more effective than simply reintroducing low-level trajectory supervision.

### 3. Latent space analysis

To support the claim that latent-space planning improves semantic modeling, we added:

- **Clustering in latent space** with decoded cluster centroids, showing distinct maneuver types (turns, braking, different speeds).
- **Trajectory clustering + UMAP** visualizations, where latent codes form compact, well-separated intent clusters and outliers correspond to “staying still”.

We also show that a VAE with lower reconstruction error can give worse planning performance, highlighting a trade-off between reconstruction fidelity and semantic compactness; early stopping yields better downstream planning.

### 4. Stability, significance & OOD robustness

To address concerns on variance and OOD behavior:

- All LAP scores are now **averaged over 5 closed-loop runs**; even the worst LAP run on Test14-hard (NR) remains ≈2 points above Diffusion Planner.
- Re-running the full VAE–diffusion training with three additional seeds yields <1% variation across splits, indicating a stable pipeline.
- We clarify that we use the same trajectory-level data augmentations as prior work and that the more compact, early-stopped VAE generalizes better to OOD semantics.

### 5. Single-step inference vs. multi-modality

We quantify the trade-off between speed and diversity:

| Variant        | APD ↑ | FPD ↑ |
|:---------------|:-----:|:-----:|
| LAP (10 steps) | 2.03  | 4.55  |
| LAP (1 step)   | 0.10  | 0.22  |

Single-step decoding is thus presented as a **latency-critical mode** (near-mean trajectory), while multi-step decoding fully preserves the multi-modal nature of LAP.


More detailed responses can be found in our individual comments to the reviewers. All changes have been incorporated into the [revised version](https://openreview.net/pdf?id=uHEaVkj8I3) and highlighted in blue.

---

### Meta-Review · Area_Chair_8BhV · 2026-01-05

**Summary:**

The paper proposes LAP, a two-stage planning framework that compresses trajectories into a latent space via a VAE before applying a diffusion model. To preserve map-level details, the architecture incorporates a "feature distillation" module. Initial scores are mixed (4, 4, 6).

I recommend rejection. While the reported 10x inference speed-up is notable, the proposed "feature distillation" module adds significant engineering complexity. Ultimately, the marginal performance gains on the validation split do not justify this architectural overhead.

**Reviewer Concerns:**

Addressed Concerns:

- Interpretability of Latent Space (Reviewer iCPG, MsQH): Reviewers questioned whether the latent representations captured semantic meaning. Through new visualizations (Clustering and UMAP), the authors demonstrated that the space effectively disentangles high-level intents (e.g., turning vs. stopping) rather than performing arbitrary compression.
- Source of Efficiency (Reviewer dMwV): The authors clarified that the efficiency gains stem from reducing sampling steps (from 20 to 2) rather than a reduction in GFLOPs.

Outstanding Concerns:

- Utility of Distillation Module (Reviewer dMwV, MsQH): The ablation study proved the module is *necessary* for performance, but it didn't convince Reviewer dMwV that the design itself is good. The 'complexity' concern stands.

- Diversity vs. Speed Trade-off (Reviewer MsQH): Concerns were raised regarding the impact of single-step inference on multi-modality. The authors acknowledged a trade-off: the diversity metric (APD) drops from 2.03 to 0.10 in single-step mode, indicating a collapse towards a mean trajectory.
- Robustness of Gains (Reviewer iCPG): While the authors report a +3.08 gain on Test14-hard, the model shows negligible improvement or slight degradation on the *Val14* split. Consequently, the reviewer remains skeptical about the robustness of the SOTA claims.

**Reviewer Scores:**

- Reviewer MsQH (Initial: 6): 6 (Maintain). Appreciates the architecture and SOTA results. This reviewer did not respond to the rebuttal, but the authors provided the requested implementation details.
- Reviewer dMwV (Initial: 4): 4 (Maintain). Primarily a design disagreement. They view the distillation module as "complicated and seems unnecessary".
- Reviewer iCPG (Initial: 4): 4 (Maintain). Remains skeptical about the "black box" nature of the latent space and suspects the performance gains might partially be due to simulator noise, despite the new visualizations.

---

### Decision · Program_Chairs · 2026-01-26

Reject